# Archetypal SAE: Adaptive and Stable Dictionary Learning for Concept Extraction in Large Vision Models

**Thomas Fel** [* 1]  **Ekdeep Singh Lubana** [* 2]  **Jacob S. Prince** [3]  **Matthew Kowal** [4 5]  **Victor Boutin** [6]
**Isabel Papadimitriou** [1]  **Binxu Wang** [1]  **Martin Wattenberg** [7 8]  **Demba Ba** [1]  **Talia Konkle** [9]

## Abstract

Sparse Autoencoders (SAEs) have emerged as a powerful framework for machine learning interpretability, enabling the unsupervised decomposition of model representations into a dictionary of abstract, human-interpretable concepts. However, we reveal a fundamental limitation: existing SAEs exhibit severe instability, as identical models trained on similar datasets can produce sharply different dictionaries, undermining their reliability as an interpretability tool. To address this issue, we draw inspiration from the Archetypal Analysis framework introduced by Cutler & Breiman (1994) and present Archetypal SAEs (**A-SAE**), wherein dictionary atoms are constrained to the convex hull of data. This geometric anchoring significantly enhances the stability of inferred dictionaries, and their mildly relaxed variants **RA-SAE**s further match state-of-the-art reconstruction abilities. To rigorously assess dictionary quality learned by SAEs, we introduce two new benchmarks that test (*i*) plausibility, if dictionaries recover "true" classification directions and (*ii*) identifiability, if dictionaries disentangle synthetic concept mixtures. Across all evaluations, **RA-SAE**s consistently yield more structured representations while uncovering novel, semantically meaningful concepts in large-scale vision models.

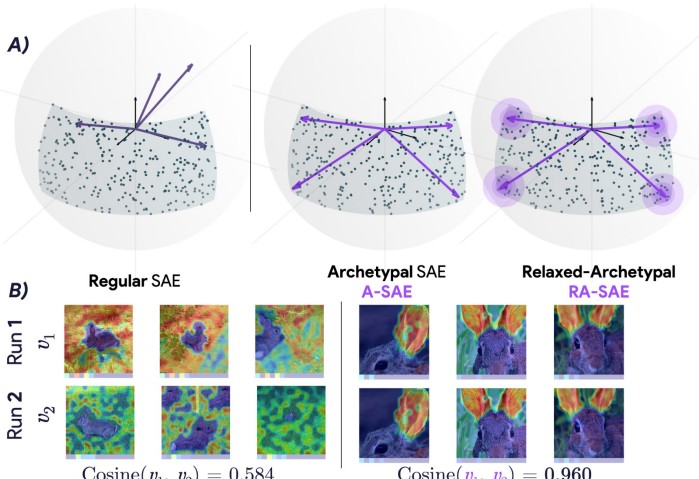

*Figure 1.* **A) Archetypal-SAE.** Archetypal-SAEs constrain dictionary atoms (decoder directions) to the data's convex hull, improving stability. A relaxed variant (**RA-SAE**) allows mild relaxation, matching standard SAEs in reconstruction while maintaining stability. Both integrate with any SAE variant (e.g., TopK, JumpReLU). **B) Instability Problem.** Standard SAEs produce inconsistent dictionaries across runs, undermining interpretability. For example, in classical SAEs, the second most important concept for "rabbit" in one run has no counterpart in another run ($\cos = 0.58$). In contrast, Archetypal-SAEs maintain consistent concept correspondences across runs, ensuring stability.

---

[*]Equal contribution  [1]Kempner Institute, Harvard University [2]CBS-NTT Program in Physics of Intelligence, Harvard University [3]Dept. of Psychology, Harvard University [4]FAR.AI [5]York University [6]CerCo, CNRS [7]Harvard University [8]Google [9]Dept. of Psychology & Kempner Institute, Harvard University. Correspondence to: Thomas Fel <tfel@g.harvard.edu>.

*Proceedings of the 42nd International Conference on Machine Learning*, Vancouver, Canada. PMLR 267, 2025. Copyright 2025 by the author(s).

# 1. Introduction

Artificial Neural Networks (ANNs) have revolutionized computer vision, setting new benchmarks across a wide range of tasks. Despite these successes, the "black-box" nature of ANNs poses significant challenges, particularly in domains requiring transparency, accountability, and adherence to strict ethical and regulatory standards (Tripicchio & D'Avella, 2020). Beyond compliance, there is a growing curiosity within the scientific community to better understand these models' internal mechanisms, both to satisfy fundamental questions about their function, leverage insights for debugging (Adebayo et al., 2020) and improvement, and even explore parallels with neuroscience (Goodwin et al., 2022; Vilas et al., 2024). These motivations have spurred the rapid growth of explainable artificial intelligence (XAI), a field dedicated to enhancing the interpretability of ANNs, thereby bridging the gap between machine intelligence and human understanding (Doshi-Velez & Kim, 2017).

Among XAI approaches, concept-based methods (Kim et al., 2018) have emerged as a powerful framework for uncovering intelligible visual concepts embedded within the intricate activation patterns of ANNs (Ghorbani et al., 2019; Zhang et al., 2021; Fel et al., 2023b; Graziani et al., 2023; Vielhaben et al., 2023). These methods excel in making the internal representations of ANNs more comprehensible by associating them with human-interpretable concepts. Recently, concept extraction methods have been shown to be instances of dictionary learning (Fel et al., 2023a), where the goal is to map the activation space of an ANN into a (higher-dimensional), sparse "concept space". The resulting concept basis is often considered more interpretable. For such representations to be effective, they must be as sparse as possible while still enabling accurate reconstruction of the original activations from the learned basis using a dictionary of atoms—also called prototypes. Historically, dictionary learning methods have included techniques such as Non-negative Matrix Factorization (NMF) (Lee & Seung, 2001; Kowal et al., 2024a; Jourdan et al., 2023) and K-Means (Gersho & Gray, 1991; Ghorbani et al., 2019), while more recent approaches like Sparse Autoencoders (SAEs) (Cunningham et al., 2023; Bricken et al., 2023; Rajamanoharan et al., 2024; Gao et al., 2025; Thasarathan et al., 2025; Poché et al., 2025) have emerged as a powerful alternative. SAEs achieve a good balance between sparsity and reconstruction quality, and their optimization frameworks scale well to large datasets. However, compared to traditional methods, SAEs suffer from a critical limitation: instability. As illustrated in Figure 1, training two identical SAEs on the same data can yield significantly different dictionaries (concept bases), undermining their reliability and interpretability.

In this work, we address the instability of current SAEs

by introducing two novel variants: Archetypal-SAE (A-SAE) and its relaxed counterpart (RA-SAE). Building upon archetypal analysis (Cutler & Breiman, 1994), A-SAE constrains each dictionary atom to reside within the convex hull of the training data, ensuring a more stable and consistent set of basis elements across different training runs by virtue of this geometrical constraint. RA-SAE further extend this framework by incorporating a small relaxation term, allowing for slight deviations from the convex hull to enhance modeling flexibility while maintaining stability. Overall, our work makes the following contributions.

1. **Instability in SAEs.** We identify a critical limitation of current SAEs paradigms: two training runs on identical data can yield concept dictionaries that are largely distinct, hence compromising their reliability as an interpretability protocol.

2. **A-SAE: Archetypal anchoring to overcome instability.** To address the challenge above, we take inspiration from Cutler & Breiman (1994)'s Archetypal analysis of dictionary learning and propose **A-SAE**, an SAE paradigm wherein the dictionary atoms (decoder directions) are forced to lie in the convex hull of sample representations. As we show, this geometrical anchoring yields substantial stability across training runs. Moreover, we show a mild relaxation of this protocol, which we title **RA-SAE**, uncovers meaningful and interpretable concepts in large-scale vision models.

3. **Rigorous evaluations with novel metrics and benchmarks.** We introduce novel metrics to evaluate the quality of dictionaries learned using different SAE paradigms, while proposing two new benchmarks for testing SAEs' ability to recover classification directions and to disentangle synthetic image mixtures, inspired by identifiability theory (Locatello et al., 2019; 2020; Higgins et al., 2017). Our results provide substantial evidence that **A-SAE**s find more structured and coherent concepts. Further, to enable reproduction, we open-source our extensive codebase for large-scale SAE training on modern vision models.

# 2. Related Work

**Sparse Coding & Dictionary Learning.** Dictionary learning (Tošić & Frossard, 2011; Rubinstein et al., 2010; Elad, 2010; Mairal et al., 2014; Dumitrescu & Irofti, 2018) emerged as a fundamental approach for uncovering latent factors of a data-generating process in signal processing and machine learning, building upon early work in sparse coding (Olshausen & Field, 1996; 1997; Lee et al., 2006; Foldiak & Endres, 2008; Rentzeperis et al., 2023). The primary objective of these methods is to find a *sparse* representation of input data (Hurley & Rickard, 2009; Eamaz et al., 2022), such that each data sample can be accurately

reconstructed using a linear combination of only a small subset of dictionary atoms. The field gained momentum with compressed sensing theory (Donoho, 2006; Candès et al., 2006; Candès & Wakin, 2008; Lopes, 2013; Rencker et al., 2019), which established theoretical foundations for sparse signal recovery. Early dictionary learning methods evolved from vector quantization and K-means clustering (Gersho & Gray, 1991; Lloyd, 1982), leading to more sophisticated approaches like Non-negative Matrix Factorization (NMF) and its variants (Lee & Seung, 1999; 2001; Gillis, 2020; Ding et al., 2008; Kersting et al., 2010; Thurau et al., 2009; Gillis & Kumar, 2015), Sparse PCA, (d'Aspremont et al., 2004; Zou et al., 2006) and K-SVD (Aharon et al., 2006a; Elad & Aharon, 2006). The field further expanded rapidly (Wright et al., 2010; Chen et al., 2021; Tasissa et al., 2023) with online methods (Mairal et al., 2009; Kasiviswanathan et al., 2012; Lu et al., 2013) and structured sparsity (Jenatton et al., 2010; Bach et al., 2012; Sun et al., 2014). Theoretical guarantees for dictionary learning emerged (Aharon et al., 2006b; Spielman et al., 2012; Hillar & Sommer, 2015; Fu et al., 2018; Barbier & Macris, 2022; Hu & Huang, 2023), alongside connections to deep learning (Baccouche et al., 2012; Tariyal et al., 2016; Papyan et al., 2017; Mahdizade-haghdam et al., 2019; Tamkin et al., 2023; Yu et al., 2023). Parallel developments in archetypal analysis from Cutler & Breiman (1994) provided complementary perspectives on dictionary learning by focusing on extreme points in a set of observations (Dubins, 1962).

**Vision Explainability.** Early work in the field of Explainable AI primarily revolved around attribution methods, which highlight the input regions that most influence a model's prediction (Simonyan et al., 2013; Zeiler & Fergus, 2014; Bach et al., 2015; Springenberg et al., 2014; Smilkov et al., 2017; Sundararajan et al., 2017; Selvaraju et al., 2017; Fong & Vedaldi, 2017; Fel et al., 2021; Novello et al., 2022; Muzellec et al., 2024)—in other words, *where* the network focuses its attention to produce its prediction. While valuable, attribution methods exhibit two core limitations: (***i***) they provide limited information about the semantic organization of learned representations (Hase & Bansal, 2020; Hsieh et al., 2021; Nguyen et al., 2021; Colin et al., 2021; Kim et al., 2022; Sixt et al., 2020), and (***ii***) they can produce incorrect explanations (Adebayo et al., 2018; Ghorbani et al., 2017; Slack et al., 2021; Sturmfels et al., 2020; Hsieh et al., 2021; Hase et al., 2021). In other words, just because the explanations make sense to humans, we cannot conclude that they accurately reflect what is actually happening within the model—as shown by ongoing efforts to develop robust evaluation metrics for explainability (Petsiuk et al., 2018; Bhatt et al., 2020; Jacovi & Goldberg, 2020; Hedström et al., 2022; Fel et al., 2022; Hsieh et al., 2021; Boopathy et al., 2020; Lin et al., 2019; Idrissi et al., 2021).

To overcome the constraints above, concept-based inter-pretability (Kim et al., 2018) has gained traction. Its central objective is to pinpoint semantically meaningful directions—revealing not just *where* the model is looking, but also *what* patterns or concepts it employs—and to link these systematically to latent activations (Bau et al., 2017; Ghorbani et al., 2019; Zhang et al., 2021; Fel et al., 2023b; Graziani et al., 2023; Vielhaben et al., 2023; Kowal et al., 2024a;b). In line with this perspective, Fel et al. (2023a) demonstrate that popular concept-extraction methods : ACE (Ghorbani et al., 2017), ICE (Zhang et al., 2021), CRAFT (Fel et al., 2023b) and more recently SAEs (Cunningham et al., 2023; Bricken et al., 2023; Rajamanoharan et al., 2024; Gao et al., 2025; Surkov et al., 2024; Gorton, 2024; Bhalla et al., 2024a) all address essentially the same dictionary learning task, albeit subject to distinct constraints (see Eq. 1).

Within this broader context, we note Sparse Autoencoders (SAEs) have emerged as a highly scalable special case of dictionary learning: unlike NMF, Sparse-PCA, or other optimization problem, SAEs can be trained with the same algorithms and architectures used in modern deep learning pipelines, making them especially well-suited for large-scale concept extraction. However, motivated by more ambitious use-cases of interpretability, e.g., to develop transparency and accountability (Anwar et al., 2024), recent work has started to demonstrate limitations in existing SAE frameworks and propose improvements. Examples of such limitations include learning of overly specific or sensitive features (Bricken et al., 2023; Chanin et al., 2024), challenges in compositionality (Wattenberg & Viegas, 2024; Clarke et al., 2024), and limited effects of latent interventions (Bhalla et al., 2024b; Menon et al., 2024). In this paper, we aim to bring to attention an underappreciated challenge in SAEs' training –*instability* – wherein mere reruns of SAE training yield inconsistent interpretations (Figure 1).

## 3. (In)Stability of SAEs

We first establish the challenge of instability in SAEs' training. To this end, we start by defining notations and providing background on the SAEs analyzed in this work, then offering a formal measure of instability in SAE training.

**Notation.** Throughout this work, $||\cdot||_2$ and $||\cdot||_F$ represent the $\ell_2$ and Frobenius norms, respectively. We define $[n] = \{1, \ldots, n\}$, and consider a general representation learning framework where a deep learning model $\boldsymbol{f} : \mathcal{X} \to \mathcal{A}$ maps inputs from an input space $\mathcal{X}$ to a representation space $\mathcal{A} \subseteq \mathbb{R}^d$. The representation is captured as a set of $n$ points arranged in a matrix $\boldsymbol{A} \in \mathbb{R}^{n \times d}$, with $\boldsymbol{A}_i$ denoting its $i$-th row, where $i \in [n]$ and $\boldsymbol{A}_i \in \mathbb{R}^d$. For any matrix $\boldsymbol{X}$, $\boldsymbol{X} \geq 0$ denotes that all elements of $\boldsymbol{X}$ are non-negative, and $\mathcal{P}(n)$ is the set of $n \times n$ signed permutation matrices. Given $\boldsymbol{A} \in \mathbb{R}^{n \times d}$, we define $\text{cone}(\boldsymbol{A}) = \{\boldsymbol{x} \mid \boldsymbol{x} = \boldsymbol{A}\boldsymbol{v}, \ \boldsymbol{v} \in \mathbb{R}^n, \ \boldsymbol{v} \geq 0\}$, and $\text{conv}(\boldsymbol{A}) = \{\boldsymbol{x} \mid$

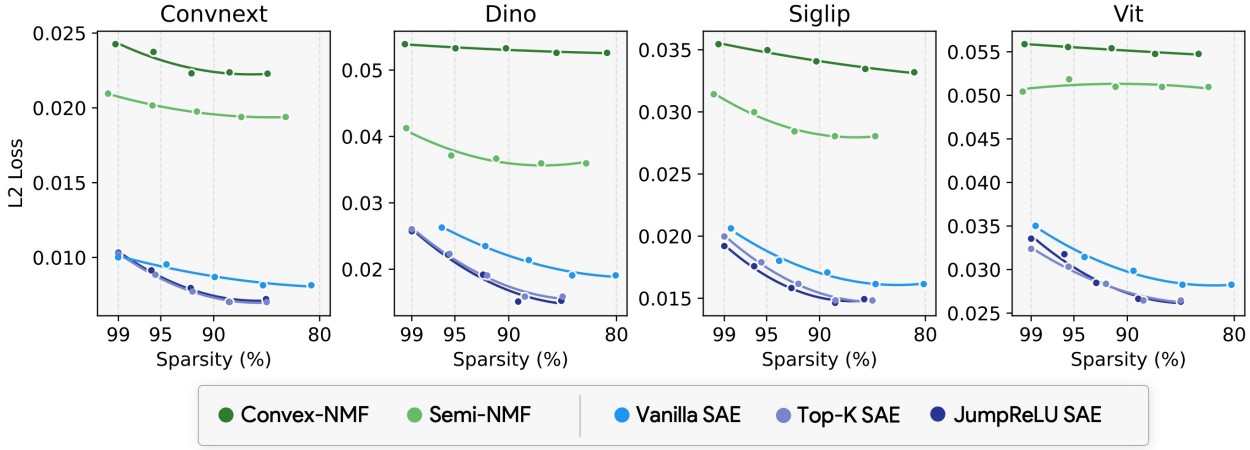

*Figure 2.* **SAEs are a promising direction for scalable concept extraction in vision.** Comparison of reconstruction error ($\ell_2$ Loss) and sparsity across four large-scale vision models: ConvNext, DINO, SigLIP, and ViT. The figure compares the performance of various dictionary learning methods, including classical approaches (Convex-NMF, Semi-NMF) and modern Sparse Autoencoders (Vanilla SAE, Top-K SAE, JumpReLU SAE). Each SAE is trained up to 250 million tokens per epoch over 50 epochs, demonstrating the scalability of SAEs and their ability to achieve superior trade-offs between reconstruction fidelity and sparsity compared to traditional methods.

$\boldsymbol{x} = \boldsymbol{A}\boldsymbol{v}, \ \boldsymbol{v} \in \mathbb{R}^n, \ \boldsymbol{v} \geq 0, \ \mathbf{1}^\top \boldsymbol{v} = 1\}$ as the conical and convex hulls, respectively, generated by the columns of $\boldsymbol{A}$.

**Concept Extraction as Dictionary Learning.** Concept extraction can be naturally framed as a dictionary learning problem, wherein a set of $n$ activations, represented by the matrix $\boldsymbol{A} \in \mathbb{R}^{n \times d}$, is approximated using a structured decomposition. The goal is to learn a **Dictionary** $\boldsymbol{D}$—also referred to as atoms (Serre, 2006), prototypes, or a codebook (Tamkin et al., 2023)—such that activations can be reconstructed as sparse linear combinations of these learned directions. The corresponding coefficients, known as **Codes** $\boldsymbol{Z}$, capture the latent structure of the activations and enforce interpretability by promoting sparsity or non-negativity. This leads to the general optimization framework:

$$(\boldsymbol{Z}^\star, \boldsymbol{D}^\star) = \operatorname*{arg\,min}_{\boldsymbol{Z}, \boldsymbol{D}} ||\boldsymbol{A} - \boldsymbol{Z}\boldsymbol{D}^\top||_F^2,$$

$$\text{s.t.} \begin{cases} \forall i, \boldsymbol{Z}_i \in \{\boldsymbol{e}_1, \ldots, \boldsymbol{e}_k\}, & (\textbf{ACE - K-Means}), \\ \boldsymbol{D}^\top \boldsymbol{D} = \mathbf{I}, & (\textbf{ICE - PCA}), \\ \boldsymbol{Z} \geq 0, \boldsymbol{D} \geq 0, & (\textbf{CRAFT - NMF}), \\ \boldsymbol{Z} = \boldsymbol{\Psi}_\theta(\boldsymbol{A}), ||\boldsymbol{Z}||_0 \leq K, & (\textbf{SAEs}). \end{cases}$$

Here, $\boldsymbol{e}_i$ denotes the $i$-th canonical basis vector, $\mathbf{I}$ is the identity matrix, and $\boldsymbol{\Psi}_\theta$ is a neural network parameterized by $\theta$ (typically a single feedforward layer with bias). Notably, in Sparse Autoencoder (SAE) literature, the dictionary $\boldsymbol{D}$ is often identified with the decoder weight matrix, denoted as $\boldsymbol{W}_{\text{dec}}$. This optimization framework unifies various classical methods for concept extraction, ranging from clustering-based approaches (Ghorbani et al., 2019), orthogonal factorization methods (Zhang et al., 2021; Graziani et al., 2021),

and nonnegative matrix factorization (Fel et al., 2023b) to modern sparse autoencoding techniques (SAEs) (Cunningham et al., 2023; Bricken et al., 2023).

Despite similar formulations to SAEs, solving K-Means, PCA, or NMF typically does not rely on backpropagation and lacks inherent batch-learning capabilities. This make SAEs particularly appealing for large-scale applications. Additionally, optimization problems solved in multiple steps, such as NMF, Semi-NMF, or K-Means, can be interpreted as having a multi-layer nonlinear encoding (akin to a $\boldsymbol{\Psi}_\theta$ with multiple layers)[1]. We note that the decoding process in these methods remains linear.

To compare these approaches, we generally evaluate the trade-off between two metrics: *sparsity*, measured as the $\ell_0$ norm of $\boldsymbol{Z}$, and *fidelity*, measured as the $\ell_2$ reconstruction error. As a starting point, we propose to study this pareto frontier using state-of-the-art SAEs, including JumpReLU (Rajamanoharan et al., 2024), TopK (Gao et al., 2025), and a vanilla SAE with $\ell_1$ regularization (Bricken et al., 2023), alongside classical sparse dictionary learning methods. Since PCA is non-sparse, K-Means sparsity is fixed, and NMF is applicable only to non-negative activations, we adopt modified versions of these methods that are broadly applicable for concept extraction (Kowal et al., 2024a; Parekh et al., 2024), such as Convex-NMF and Semi-NMF. For SAEs, we use the following standard formulation:

$$\boldsymbol{\Psi}_\theta(\boldsymbol{A}) = \boldsymbol{\sigma}(\boldsymbol{A}\boldsymbol{W}_\theta + \boldsymbol{b}), \qquad (1)$$

where $\boldsymbol{A}$ denotes a linear encoder layer and $\boldsymbol{\sigma}(\cdot)$ is an ele-

---

[1] A single optimization step in a NMF is akin to a linear model followed by a ReLU (projection onto the positive orthant).

mentwise non-linearity that depends on the specific SAE architecture. For all SAEs, the resulting codes $\boldsymbol{Z} = \boldsymbol{\Psi}_\theta(\boldsymbol{A}) \geq 0$ holds. We employ a Silverman kernel (Silverman, 1984) for instantiating Jump-ReLU SAEs. For vanilla SAEs, an $\ell_1$ regularization on $\boldsymbol{Z}$ is applied until the desired sparsity is achieved. Top-K and Jump-ReLU SAEs directly control or optimize an $\ell_0$ constraint.

Results in Figure 2 illustrate that the SAE methods discussed above outperform optimization based dictionary learning methods in terms of reconstruction fidelity for fixed sparsity levels. While this positions SAEs as a compelling solution for concept extraction, as we show in the following, a significant drawback of current methods lies in their instability: minor changes to the dataset can lead to substantial variations in the learned dictionary.

**Measuring Instability.** To formally define a notion of stability in SAEs' training, we seek inspiration from the dictionary learning literature and prior works (Spielman et al., 2012), yielding the following metric for two dictionaries $\boldsymbol{D} \in \mathbb{R}^{n \times d}$ and $\boldsymbol{D}'$:

$$\text{Stability}(\boldsymbol{D}, \boldsymbol{D}') = \max_{\boldsymbol{\Pi} \in \mathcal{P}(n)} \frac{1}{n} \text{Tr}(\boldsymbol{D}^\mathsf{T} \boldsymbol{\Pi} \boldsymbol{D}'), \quad (2)$$

where we assume that each atom lies on the unit $\ell_2$-norm ball, i.e., $||\boldsymbol{D}_i||_2 = 1$ for all $i \in [n]$. This metric essentially measures the optimal average cosine similarity between the dictionaries after finding the best alignment via the Hungarian algorithm[2].

We evaluate stability across various vision models by altering only the random seed of the algorithm, while keeping the dataset unchanged across 4 runs. A similar trend is observed when the dataset is perturbed by as little as 5% or 10%, consistent with prior findings (Fel et al., 2023a; Braun et al., 2024; Paulo & Belrose, 2025). Results in Figure 3 show that while SAEs outperform classical dictionary learning methods in terms of reconstruction error ($\ell_2$-loss), they exhibit lower stability, with cosine stability values around 0.5 for TopK SAE trained on DinoV2 with over 250 million tokens. As a first approximation, this implies that re-running the same algorithm with a different seed can result in dictionaries where only half the concepts remain, while the other half are new orthogonal concepts. By contrast, the proposed Archetypal SAE, introduced below, achieves stability comparable to classical dictionary learning methods without compromising reconstruction fidelity.

## 4. Towards Archetypal SAEs

In their seminal work, Cutler & Breiman (1994) proposed representing each data point as a convex combination of

*Archetypes*, which are themselves defined as convex combinations of data points. Concretely, the dictionary (the collection of Archetypes) can be constructed by multiplying the data by a row-stochastic matrix. Drawing on these ideas, we propose a solution to the problem of instability in SAEs' training: the **Archetypal SAE** (**A-SAE**), which acts as a plug-and-play parameterization of the dictionary—i.e., the decoder matrix—and can be seamlessly integrated into any existing SAE, e.g., TopK (Gao et al., 2025) or Jump-ReLU (Rajamanoharan et al., 2024).

**Formulation.** Let $\boldsymbol{A} \in \mathbb{R}^{n \times d}$ represent the data matrix (with $n$ data points in $\mathbb{R}^d$), and $\Delta^n = \{\boldsymbol{x} \in \mathbb{R}^n \mid x_i \geq 0, \mathbf{1}^\mathsf{T} \boldsymbol{x} = 1\}$ denote the $(n{-}1)$-dimensional simplex in $\mathbb{R}^n$. A matrix $\boldsymbol{W} \in \mathbb{R}^{k \times n}$ is *row-stochastic* if each row $\boldsymbol{W}_i$ belongs to $\Delta^n$. Define the set of row-stochastic matrices as

$$\Omega_{k,n} \triangleq \{\boldsymbol{W} \in \mathbb{R}^{k \times n} \mid \boldsymbol{W} \geq 0, \boldsymbol{W} \mathbf{1}_n = \mathbf{1}_k\}. \quad (3)$$

An archetypal dictionary $\boldsymbol{D}$ is then defined as follows.

$$\boldsymbol{D} = \boldsymbol{W} \boldsymbol{A}, \quad \text{where } \boldsymbol{W} \in \Omega_{k,n}. \quad (4)$$

Hence, each row of $\boldsymbol{D}$ is a convex combination of the rows of $\boldsymbol{A}$, ensuring that each archetype originates from the data.

**Geometric Interpretation.** In standard SAEs (and most dictionary learning approaches), the dictionary $\boldsymbol{D}$ is *free* in the sense that each atom $\boldsymbol{D}_i \in \mathbb{R}^d$ can be placed anywhere in the ambient space. From a geometric perspective, this flexibility allows the reconstructions $\boldsymbol{Z}\boldsymbol{D}$ to span regions that may exceed the convex hull of the data $\boldsymbol{A}$. While this unconstrained setting enables greater expressivity, it comes with significant drawbacks. Specifically, small perturbations in the data or random initializations can lead to unstable solutions, resulting in dictionaries that differ drastically across training runs. Moreover, if the dictionary atoms $\boldsymbol{D}_i$ point in directions unrelated to the data, probing these directions may fail to activate any meaningful mechanisms within the underlying model (Makelov et al., 2023). This highlights the importance of ensuring that the dictionary aligns with "real" directions inherent to the data.

In contrast to above, the *Archetypal* dictionary imposes a crucial geometric restriction: every dictionary atom is constrained to lie within the convex hull of $\boldsymbol{A}$, i.e., each dictionary atom $\boldsymbol{D}_i$ is a convex combination of samples' representations. Thus, once multiplied by a nonnegative $\boldsymbol{Z}$, the reconstructions $\boldsymbol{Z}\boldsymbol{D}$ remain within the *conic* hull of $\boldsymbol{A}$. This anchoring within the data manifold precludes the emergence of pathological or out-of-sample directions, yielding stability gains shown empirically in Figure 3 and formalized in Proposition G.2. Concretely, we always have:

$$\boldsymbol{D} \in \text{conv}(\boldsymbol{A}), \quad \boldsymbol{Z}\boldsymbol{D} \in \text{cone}(\boldsymbol{A}).$$

---

[2]In practice, we use the `linear_sum_assignment` function from `scipy` (Virtanen et al., 2020), which solves the linear sum assignment problem by finding the optimal matching in a bipartite graph using the Jonker-Volgenant algorithm.

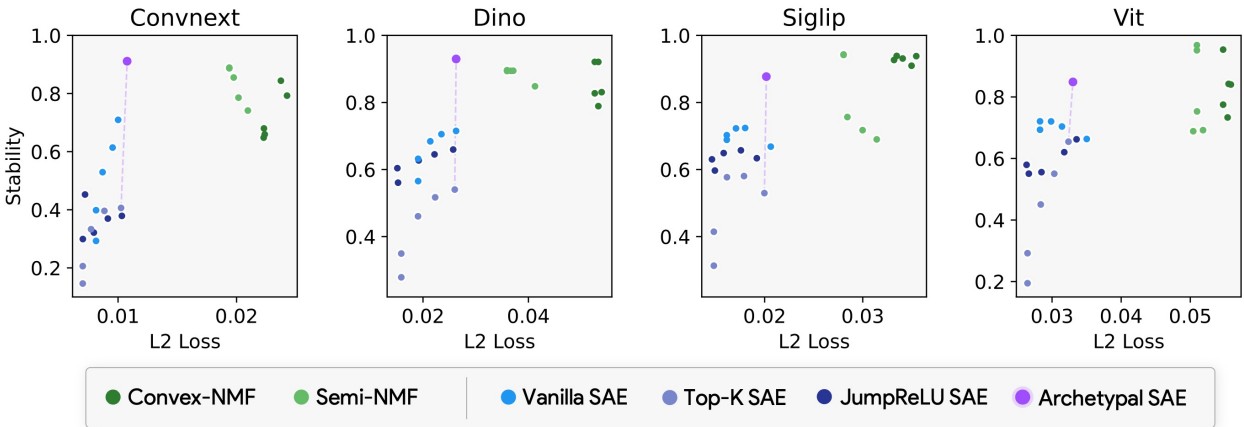

*Figure 3.* **Stability-Reconstruction tradeoff** (optimal: top-left). We implement 5 dictionary learning methods on 4 models at 5 levels of sparsity each, as well as our A-SAE method. We show that SAEs exhibit instability (minor perturbations in the dataset can lead to significant changes in the learned dictionary), while traditional dictionary learning methods are more stable but worse at reconstructing the data. Archetypal-SAEs (ours) help mitigate this issue. We measure stability based on Equation (2): the optimal average cosine similarity between the dictionaries across 4 runs after finding the best alignment via the Hungarian algorithm. Archetypal-SAEs improve stability without compromising reconstruction fidelity, performing better on the stability-reconstruction tradeoff than existing methods.

Moreover, one can restrict $D$ to be formed from a subset $C \subseteq \mathbb{R}^{n' \times d}$ (rather than all of $A$) without losing expressivity, provided $C$ contains the extreme points of $A$ (Dubins, 1962). Indeed, in that case $\text{cone}(C) = \text{cone}(A)$, ensuring the same representational power (see Proposition G.1).

We must now ask the core question that makes SAEs exciting: does an SAE with an Archetypal dictionary, as defined above, scale? Indeed, optimizing the matrix $W$ of size $k \times n$, where $n$ is the number of points and $k$ is the number of concepts, is often infeasible in practice (e.g., when the number of tokens $n > 10^8$). This limitation speaks to the importance of the matrix $C$, which can either be a subset of $A$ or elements within $\text{conv}(A)$, such as mixtures of points. Specifically, accessing extreme points is necessary to achieve perfect reconstruction (Simon, 2011), but this is also intractable in practice for such high dimensions. However, we propose to address this with a relaxation.

**Scaling Archetypal-SAE.** To maintain the desirable properties of **A-SAE** while addressing scalability, we propose using a reduced subset of points $C$, with $n' \ll n$, chosen as centroids of $A$ obtained via K-Means. We fix $C$ and train only $W$. We apply a ReLU activation and normalize each row of $W$ at each step to ensure that $W \in \Omega_{k,n}$. Experiments indicate that K-Means forms the most reliable method for distilling $A$ into $C$ (see Figure 11), compared to alternatives such as isolation forests (Liu et al., 2008), convex hull computation in reduced dimensions, or outlier detection methods (Scholkopf et al., 1999; Breunig et al., 2000) (details in Appendix E). Hereafter, we use **A-SAE** to refer to this implementation where we directly optimize $W$ to

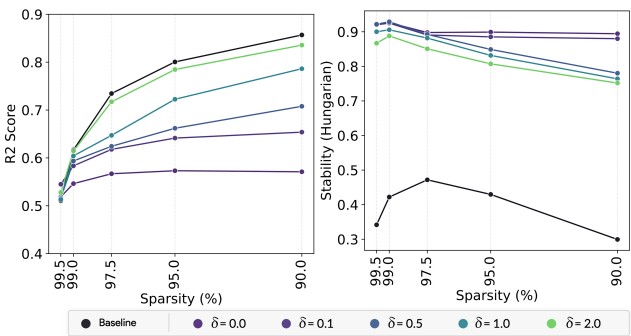

*Figure 4.* **Impact of the Relaxation Parameter** ($\delta$). Enumerating extreme points is infeasible in practice; therefore, we introduce a small relaxation parameter ($\delta$) that allows exploration beyond the convex hull of $C$. The magnitude of this relaxation enables the Archetypal SAE to achieve performance comparable to the unconstrained TopK SAE denoted as Baseline (**left**) while maintaining excellent stability (**right**).

find the best convex combination of points that reconstruct data. To enable a controlled degree of flexibility beyond $\text{conv}(C)$, we introduce a mild relaxation term $\Lambda \in \mathbb{R}^{k \times d}$, a matrix of the same dimensions as the dictionary, with a small norm constraint $||\Lambda||_2^2 \leq \delta$. This leads to a relaxed formulation, which we call the **Relaxed Archetypal SAE** (**RA-SAE**). Unlike standard A-SAEs, **RA-SAE** learns both the convex combination weights $W$ and the relaxation term $\Lambda$, while ensuring that deviations from the convex hull remain minimal. Formally, we define the dictionary as:

$$D = WC + \Lambda, \quad s.t. \quad W \in \Omega_{k,n} \text{ and } ||\Lambda||_2^2 \leq \delta.$$

Here, only $W$ and $\Lambda$ are trainable parameters, as detailed in the pseudocode (Figure 5). This implementation ensures that $W$ remains row-stochastic and that the deviation term $\Lambda$ stays within the prescribed norm constraint. As shown in Figure 4, **RA-SAE** achieves reconstruction performance on par with conventional Top-K SAEs while maintaining the stability benefits of the archetypal constraint.

```python
class ArchetypalDictionary(nn.Module):
  def __init__(self, C, k, δ=1.0):
    super().__init__()
    n', d = C.shape
    self.register_buffer("C", C)
    self.W = nn.Parameter(torch.eye(k, n'))
    self.Λ = nn.Parameter(torch.zeros(k, d))
    self.δ = δ

  def forward(self, Z):
    with torch.no_grad():
      W = torch.relu(self.W)
      W /= W.sum(dim=-1, keepdim=True)
      Λ *= torch.clamp(
          self.δ / Λ.norm(dim=-1, keepdim=True),
          max=1
      )
    D = W @ self.C + Λ
    return Z @ D
```

*Figure 5.* **Pseudocode for Relaxed Archetypal SAE (RA-SAE).** This implementation ensures that dictionary atoms remain close to convex hull of the data $\text{conv}(C)$ while allowing controlled deviations for better flexibility.

# 5. Experiments

This section is organized into five parts. We begin by describing the experimental setup in detail. Then, we introduce a novel set of theoretical metrics designed to better understand the differences between optimization-based dictionary learning methods, standard SAEs, and the proposed Archetypal SAEs. Following this, we present a new benchmark inspired by identifiability theory, which serves to evaluate the plausibility and uniqueness of the learned representations. We then propose a second, more practical benchmark to assess whether the models can retrieve "true" directions or concepts effectively utilized by the models. Finally, we conclude with qualitative examples showcasing the concepts discovered by SAEs, particularly when applied to the DINOv2 model, providing insight into the interpretability and utility of the learned representations.

**Setup.** We evaluate five models: DINOv2 (Darcet et al., 2023; Oquab et al., 2023), SigLip (Zhai et al., 2023), ViT (Dosovitskiy et al., 2020), ConvNeXt (Liu et al., 2022), and ResNet50 (He et al., 2016), sourced from the timm library (Wightman et al., 2019). Unless specified, we trained

overcomplete dictionaries with size $k = 5\times$ the feature dimension (e.g., $768 \times 5$ for DINOv2 and $2048 \times 5$ for ConvNeXt). Models were trained on the ImageNet dataset ($\sim 1.28$M images), resulting in over 60M tokens per epoch for ConvNeXt ($7 \times 7$ tokens/image) and over 250M tokens per epoch for DINOv2 ($14 \times 14$ patches/image) across 50 epochs. Semi-NMF and Convex-NMF were trained using gradient descent with accumulation and $\ell_1$ regularization to control sparsity, while the **RA-SAE** was applied atop a TopK SAE to maintain consistent sparsity. To compute $C$, we applied K-Means clustering to the entire dataset, reducing it to 32,000 centroids, which achieved reconstruction error comparable to the unconstrained SAE. The data matrix $A$ was element-wise standardized. All the SAEs were trained using the Overcomplete library[3].

## 5.1. Dictionary Learning Metrics

To improve SAEs, it is essential to deepen our understanding of the solutions they yield. We thus evaluate both standard and novel sets of metrics that evaluate dictionary learning methods across four key dimensions: (*i*) sparse reconstruction, (*ii*) consistency, (*iii*) structure in the dictionary ($D$), and (*iv*) structure in the codes ($Z$), all reported in Tab. 1 (see App. D for formal definitions). Overall, we find Archetypal SAEs achieve a strong balance between reconstruction performance, consistency, and identification of structure.

*i*) **Sparse Reconstruction.** Prior work (Bricken et al., 2023) commonly assesses the quality of the learned dictionary to reconstruct the original data under sparsity constraints via metrics such as $R^2$, sparsity ($\ell_0$ norm), and the effective usage of dictionary atoms (e.g., dead codes). We find that Archetypal SAEs perform on-par with existing SAEs and outperform NMF methods in reconstruction, achieving higher $R^2$ scores for comparable sparsity levels.

*ii*) **Consistency.** Excelling in reconstruction does not guarantee that the learned solution aligns with the underlying data distribution or is stable across training runs. To this end, we measure stability, which assesses the consistency of learned dictionaries across different initializations or perturbations in the data, and the OOD score, which quantifies how close the dictionary atoms are to real data points, i.e., whether learned concepts remain grounded and interpretable. Our findings indicate that SAEs perform poorly in consistency, showing both low stability (as evidenced in Figure 3) and suboptimal OOD scores. In contrast, Archetypal SAEs significantly enhance stability and OOD score without sacrificing reconstruction performance. In Proposition G.4, we provide theoretical arguments showing that, under mild assumptions, the OOD score of Archetypal SAEs is inherently lower-bounded, ensuring that learned dictionary atoms remain well-anchored within the data.

---

[3]https://github.com/KempnerInstitute/overcomplete

| Metric | Van. SAE | TopK SAE | Jump SAE | SNMF | CNMF | RA-SAE |
|--------|----------|----------|----------|------|------|--------|
| $R^2$ ($\uparrow$) | 83.94 | _89.52_ | **89.92** | 67.43 | 55.48 | 89.34 |
| Dead Codes ($\downarrow$) | **0.00** | **0.00** | **0.00** | 0.064 | 0.031 | 0.02 |
| Stability ($\uparrow$) | 0.710 | 0.542 | 0.539 | 0.925 | **0.933** | _0.927_ |
| Max Cosine ($\uparrow$) | 0.997 | 0.993 | 0.994 | **0.999** | **0.999** | **0.999** |
| OOD Score ($\downarrow$) | 0.451 | 0.551 | 0.551 | 0.430 | _0.087_ | **0.060** |
| Stable Rank ($\downarrow$) | 86.8 | 141.6 | 128.0 | **5.38** | 6.65 | _5.89_ |
| Eff. Rank ($\downarrow$) | 363 | 372 | 371 | **186** | _289_ | 310 |
| Coherence ($\downarrow$) | 0.838 | _0.728_ | **0.560** | 0.999 | 0.999 | 0.973 |
| Connectivity ($\uparrow$) | 0.000 | 0.002 | 0.003 | **0.243** | 0.138 | _0.159_ |
| Neg. Inter. ($\downarrow$) | 39.99 | 135.7 | 243 | _0.005_ | **0.002** | 0.012 |

*Table 1.* **Quantitative comparison of the dictionary learning methods on DINOv2,** using a 90% sparse, overcomplete dictionary with 2000 concepts. SAE methods achieve the highest reconstruction performance. C-NMF, S-NMF, and Archetypal methods excel in consistency, ensuring stability across runs and that found concepts are close to real data (OOD). Additionally, these methods demonstrate superior dictionary structure (Stable and Eff. Rank) and codes structure (Connectivity and Neg Inter.), indicating patterns in the inferred concepts and structured sparsity.

***iii*) Structure in the Dictionary ($D$).** This dimension examines whether the learned dictionary exhibits meaningful patterns. A well-structured dictionary may reveal meta-concepts or decomposable higher-level abstractions. Metrics such as stable rank, effective rank, and coherence provide insights into the compactness and interpretability of the dictionary. In particular, among solutions with comparable reconstruction performance, a more structured dictionary is preferable due to its potential for higher interpretability and organization. Across runs, methods like CNMF, SNMF, and Archetypal dictionary learning consistently yield dictionaries with better structure, reflecting their capacity for capturing higher-level patterns within the data. We also provide theoretical arguments in Proposition G.3, demonstrating that the rank of Archetypal dictionaries is inherently bounded by the rank of the data matrix.

***iv*) Structure in the Codes ($Z$).** The structure in the encoding space is equally important, as it determines how concepts are combined to reconstruct the data. Connectivity, measured as the $\ell_0$ norm of $ZZ^\top$, reflects the combinatorial diversity of the concepts. High connectivity enables complex reconstructions, while low connectivity highlights structural sparsity and simpler patterns. Additionally, negative interference (Neg. Inter.) quantifies the simultaneous activation of conflicting concepts, which can cancel each other out. Archetypal SAEs and optimization-based dictionary learning approaches reliably produce more structured codes with reduced interference, enhancing the coherence of their representations.

### 5.2. Plausibility Benchmark

We evaluate SAEs' ability to recover true classification directions by assessing whether the learned dictionary $D$ aligns with the classifier's final layer weights $\{v_1, \ldots, v_c\}$, where $c$ is the number of classes. Specifically, for each class vector $v_i$, we compute the most aligned dictionary

| Dict. size (k) | 512 | 1k | 2k | 4k | 8k | 16k | 32k |
|----------------|-----|-----|-----|-----|-----|------|------|
| **ConvNeXt** | | | | | | | |
| Baseline (TopK SAE) | 0.1681 | 0.1686 | 0.1668 | 0.1671 | 0.1684 | 0.1692 | 0.1684 |
| A-SAE ($\delta = 0$) | 0.2172 | 0.3046 | 0.3597 | 0.3887 | 0.3957 | 0.3984 | 0.3999 |
| RA-SAE ($\delta = 0.01$) | 0.1973 | 0.2887 | 0.3581 | 0.3900 | 0.4007 | 0.4038 | 0.4045 |
| RA-SAE ($\delta = 0.1$) | 0.1270 | 0.1596 | 0.2106 | 0.2681 | 0.3280 | 0.3674 | 0.3845 |
| RA-SAE ($\delta = 1.0$) | 0.1172 | 0.1475 | 0.2124 | 0.3116 | 0.4342 | 0.5203 | 0.5581 |
| **ResNet** | | | | | | | |
| Baseline (TopK SAE) | 0.2295 | 0.2484 | 0.3055 | 0.3203 | 0.3301 | 0.3014 | 0.3125 |
| A-SAE ($\delta = 0$) | 0.5920 | 0.5985 | 0.5992 | 0.6013 | 0.6029 | 0.6105 | 0.6133 |
| RA-SAE ($\delta = 0.01$) | 0.5905 | 0.5985 | 0.5991 | 0.6013 | 0.6039 | 0.6106 | 0.6136 |
| RA-SAE ($\delta = 0.1$) | 0.5777 | 0.5932 | 0.6002 | 0.6039 | 0.6046 | 0.6067 | 0.6083 |
| RA-SAE ($\delta = 1.0$) | 0.6151 | 0.6165 | 0.6173 | 0.6189 | 0.6200 | 0.6208 | 0.6213 |
| **ViT** | | | | | | | |
| Baseline (TopK SAE) | 0.1317 | 0.1589 | 0.1984 | 0.2265 | 0.2595 | 0.2807 | 0.2939 |
| A-SAE ($\delta = 0$) | 0.2079 | 0.2459 | 0.2820 | 0.3096 | 0.3361 | 0.3581 | 0.3721 |
| RA-SAE ($\delta = 0.01$) | 0.2102 | 0.2490 | 0.2861 | 0.3132 | 0.3200 | 0.3642 | 0.3821 |
| RA-SAE ($\delta = 0.1$) | 0.2278 | 0.2747 | 0.3153 | 0.3410 | 0.3496 | 0.3912 | 0.4103 |
| RA-SAE ($\delta = 1.0$) | 0.1497 | 0.2123 | 0.2936 | 0.3624 | 0.4277 | 0.4786 | 0.5014 |

*Table 2.* **Plausibility Benchmark Results.** The Plausibility Score measures the alignment between the learned dictionary concepts and the classification head's directions. **RA-SAE** achieves a significantly higher score compared to a TopK SAE (baseline). Best scores are wrapped in **green**, and worst scores are wrapped in **red**.

atom $D_j$ and average the alignment score: Plausibility $= \frac{1}{c} \sum_{i=1}^{c} \max_j \langle v_i, D_j \rangle$. A score of 1 indicates perfect alignment, while a score of 0 implies that all concepts lie in the classifier's null space, making probing ineffective. This metric could help us detect potential hallucinations if concepts diverge too much from true classification directions, and is broadly similar to Karvonen et al. (2024); Mayne et al. (2024)'s evaluation of whether SAEs infer known linear features in toy settings. Results are reported in Table 2. We find that classical SAEs, even with extremely large dictionaries, achieve limited alignment with the classification directions (inline with prior work). In contrast, **RA-SAE** significantly enhances this alignment, recovering a substantial portion of the true classification directions. This demonstrates **RA-SAE**'s efficacy in producing semantically meaningful dictionaries.

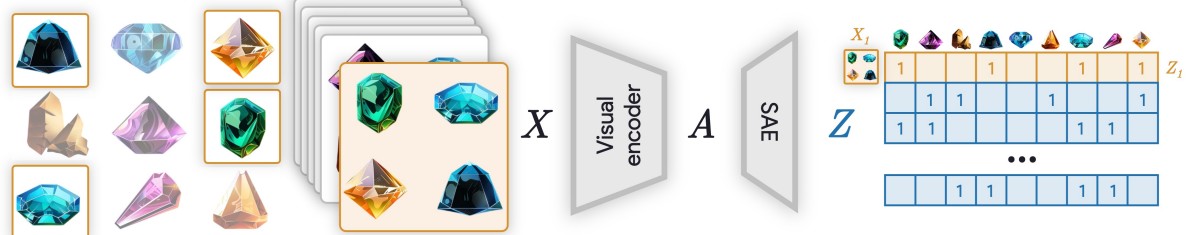

**1.** Images $(X)$ are generated from a ground truth generating process involving 9 objects forming a hidden dictionary. The model never directly observes this dictionary but only sees the resulting images.

**2.** The visual encoder extracts meaningful internal representations $(A)$ from the input images.

**3.** SAEs processes the activations $(A)$ to produce codes $Z$, containing 9 distinct concepts for each images. The ideal code **identifies** the true structure of the generative process, disentangling $A$ into interpretable components.

*Figure 6.* **Soft Identifiability benchmark.** This example uses the "gems" dataset, part of the 12 identifiability benchmarks we introduce. The goal is to evaluate whether SAEs (or any dictionary learning method under study) can disentangle and recover each object from the hidden ground truth generative process. By analyzing the model's ability to assign distinct codes to the underlying concepts, we test its capacity to reconstruct and interpret the true structure of the data.

## 5.3. Soft Identifiability Benchmark

Recent work on disentangled representation learning often evaluates whether an autoencoder trained to reconstruct samples from a data-generating process learns to represent its underlying concepts (Locatello et al., 2019; 2020; Von Kugelgen et al., 2021; Gresele et al., 2020; Khemakhem et al., 2020; Schott et al., 2021; Zimmermann et al., 2021; Menon et al., 2024). Identifiability theorems on the topic (Locatello et al., 2019; 2020) have however shown that unless the autoencoding architecture possesses the right inductive biases that match the generative process, there is no guarantee concepts underlying the data will map onto the autoencoder's latents. Since these results do not make assumptions about the data modality, they remain valid for a standard SAE training setup, e.g., similar to our experiments above. Then, an intriguing experiment involves evaluating whether when trained on representations of samples from a toy data-generating process with predefined concepts, **A-SAE**, or any other SAE architecture, develops latents capturing said concepts—if it does, then that is strongly suggestive of the SAE possessing the right inductive biases, i.e., it captures the mechanism via which a model encodes concepts in its representations.

Motivated by the above, we propose a Soft Identifiability Benchmark. Specifically, we construct twelve synthetic datasets, each comprising images formed by collaging four distinct objects (e.g., different types of gems) sampled from a pre-defined set. Each dataset is processed through a pre-trained vision model to obtain pooled activations. Ideally, when trained on these activations, the SAE is able to recover the original objects as distinct concepts within its dictionary. We then assess performance by checking whether each object class has a corresponding concept that activates above a threshold $\lambda$ when an object $y_j$ appears in the image. Formally, for each image, we feed it into a model and then into the trained SAE to get a concept-label pair $(\boldsymbol{z}, \boldsymbol{y})$, where

$\boldsymbol{z} \in \mathbb{R}^k$ represents the $k$ concept values and $\boldsymbol{y} \in \mathbb{R}^c$ denotes the $c$ class labels. We then define the accuracy for the class $j$ as: $\text{Accuracy}_j = \max_{\lambda \in \mathbb{R}, i \in [k]} \ \mathbb{P}_{(\boldsymbol{z}, \boldsymbol{y})}((z_i > \lambda) = y_j)$.

## Impact Statement

We identify a fundamental limitation of Sparse Autoencoders (SAEs): their instability undermines reliability as interpretability tools. To address this, we introduce Archetypal SAEs (**A-SAE**), which constrain dictionary atoms to the data's convex hull, and a relaxed variant (**RA-SAE**) that preserves expressivity while enhancing stability. Our evaluations, grounded in identifiability theory, demonstrate that these constraints yield more consistent, semantically meaningful, and better-structured dictionaries without sacrificing reconstruction quality.

These contributions not only improve the scientific rigor and reproducibility of SAE-based interpretability but also enable stable concept discovery at scale in large vision models. Moreover, the method generalizes across modalities, offering a robust foundation for interpretable representation learning in language models and beyond.

## Acknowledgments

This work has been made possible in part by a gift from the Chan Zuckerberg Initiative Foundation to establish the Kempner Institute for the Study of Natural and Artificial Intelligence at Harvard University. MW acknowledges support from a Superalignment Fast Grant from OpenAI, Effective Ventures Foundation, Effektiv Spenden Schweiz, and the Open Philanthropy Project.

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

# Appendix

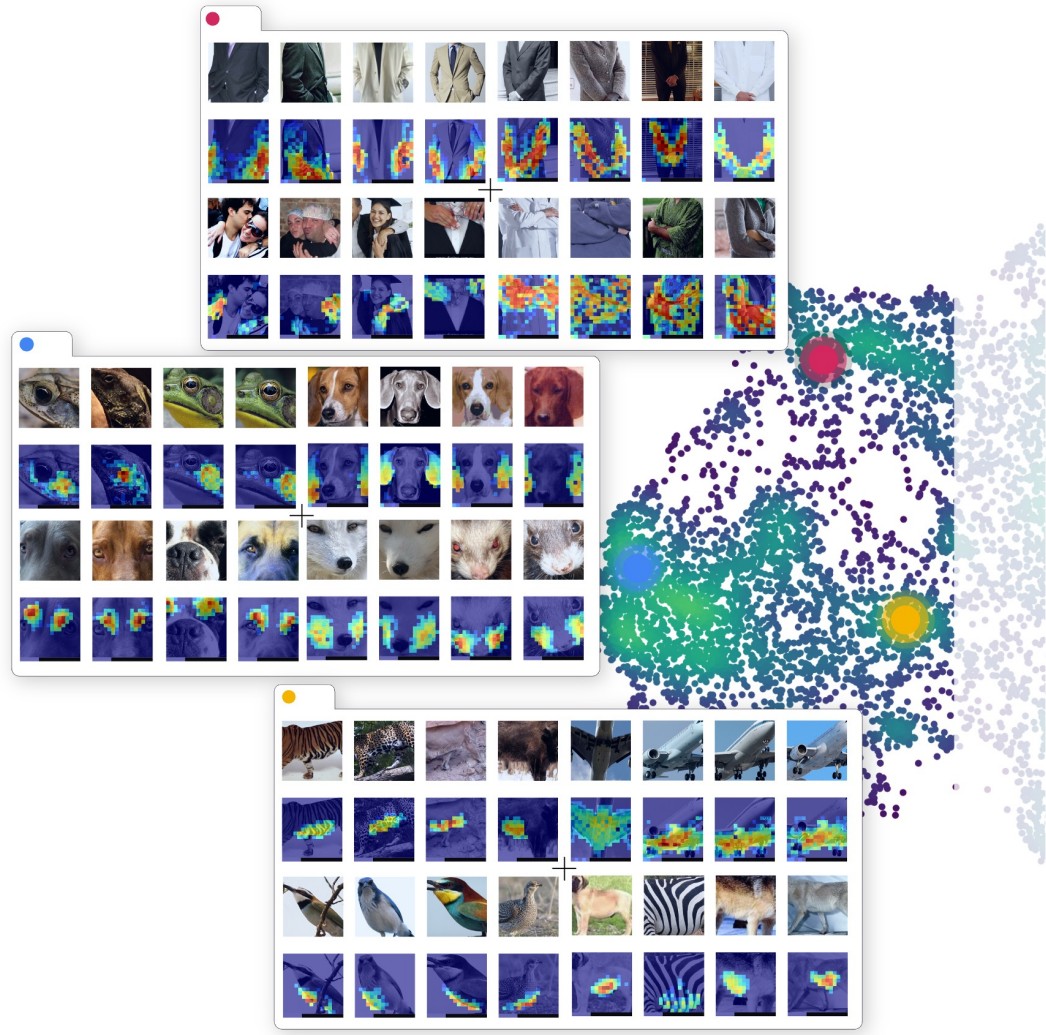

*Figure 7.* **Examples of 3 Concept Clusters in DinoV2.** Each cluster contains 4 example concepts. ● Complex hand positions, ranging from hands in pockets to hands on another person. ● Abstract "under" concepts, linking animals and objects, such as birds, zebras, felines, and airplanes, while focusing on lower regions. ● Fine-grained animal facial features, including ears, eyebrows, and cheeks.

| Method | DINOv2 | ResNet | SigLIP | ViT |
|--------|--------|--------|--------|------|
| KMeans | 0.7678 | 0.7624 | 0.7684 | 0.7702 |
| ICA | 0.8092 | 0.8370 | 0.8243 | 0.8267 |
| Sparse PCA | 0.7981 | 0.8318 | 0.8069 | 0.8082 |
| SVD | 0.7979 | 0.8291 | 0.8062 | 0.8075 |
| SemiNMF | 0.8297 | 0.8327 | 0.8358 | 0.8423 |
| ConvexNMF | 0.7645 | 0.7582 | 0.7639 | 0.7634 |
| PCA | 0.7979 | 0.8291 | 0.8062 | 0.8075 |
| Vanilla | 0.8047 | 0.8167 | 0.8126 | 0.8223 |
| TopK | 0.8135 | 0.8150 | 0.8289 | 0.8328 |
| Jump | 0.8010 | 0.7988 | 0.8131 | 0.8053 |
| A-SAE | **0.9482** | **0.9631** | **0.9602** | **0.9615** |
| RA-SAE | 0.9447 | 0.9602 | 0.9585 | 0.9586 |

*Table 3.* **Soft Identifiability benchmark Across Models and Methods.** This table presents the average accuracy scores for various methods evaluated across four different models: DINO, ResNet, SigLIP, and ViT. Best-performing scores for each model are in **bold** and second best are underlined. Full results are available in Appendix I.

## A. Soft-Identifiability results

Intuitively, the evaluation above gauges whether each object is disentangled into its "own box", in that a threshold $\lambda$ enables one-hot classification for all objects. In each of the 12 synthetic datasets, the size of the dictionary is set to the exact number of distinct objects (from 9 to 20 objects) and we generate datasets of 2,000 samples. An example of the dataset and the procedure is given in Figure 6.

Results are shown in Table 3. Classical dictionary learning methods and standard SAEs exhibit comparable performance, struggling to accurately recover the true generative factors. In contrast, **RA-SAE** consistently achieves higher accuracy in recovering the underlying object classes across all synthetic datasets. This indicates that **RA-SAE** not only stabilizes the dictionary, but also enhances its ability to discern the true underlying structure of the data, thereby improving the disentanglement of mixed objects.

## B. Qualitative Examples

This section provides qualitative insights into the concepts learned by DinoV2 (Oquab et al., 2023; Darcet et al., 2023). We trained an Archetypal SAE on DinoV2-B with 4 registers and reported key performance metrics in Figure 10. **RA-SAE** uncovers unexpected concepts, such as shadow-based features (potentially linked to depth reasoning), a context-dependent "barber" concept (activating for barbers but not clients), and fine-grained edge detection in flower petals (Figure 8). It also learns more structured within-class distinctions (e.g., separating rabbit ears, faces, and paws) compared to TopK-SAEs (Figure 9). Finally, its dictionary forms clear clusters, grouping se-

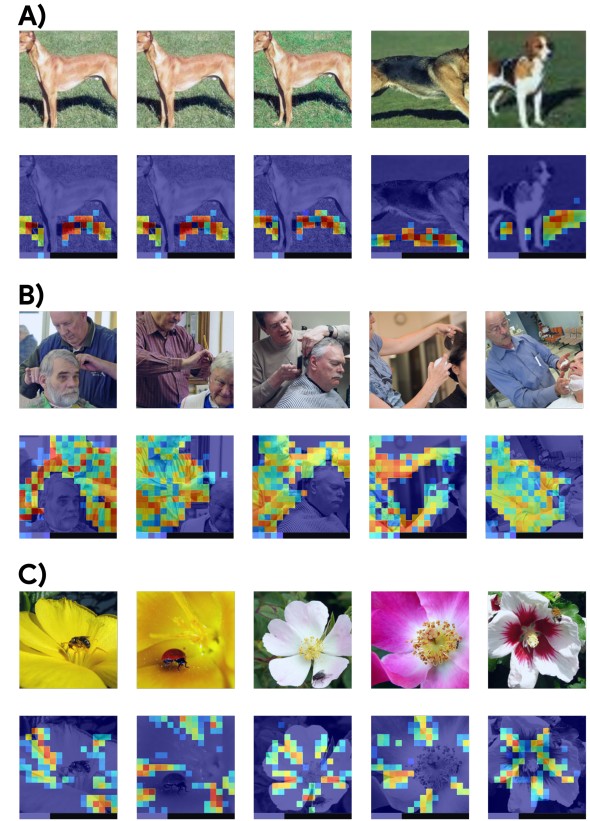

*Figure 8.* **Exotic Concepts in DinoV2. A)** Highlights tokens in shadows of dogs, suggesting shadow-based features and potential depth reasoning. **B)** A "barber" concept exclusively active for barbers, not their clients. **C)** A fine-grained concept focusing on petal edges or contours.

mantically related features like fine-grained animal faces or spatial concepts such as "underneath" (Figure 7).

To explore the dictionary learned by the model, we analyzed three clusters of concepts, as illustrated in Figure 7. Specifically:

**Baseline** - Rabbit                    **RA-SAE** - Rabbit

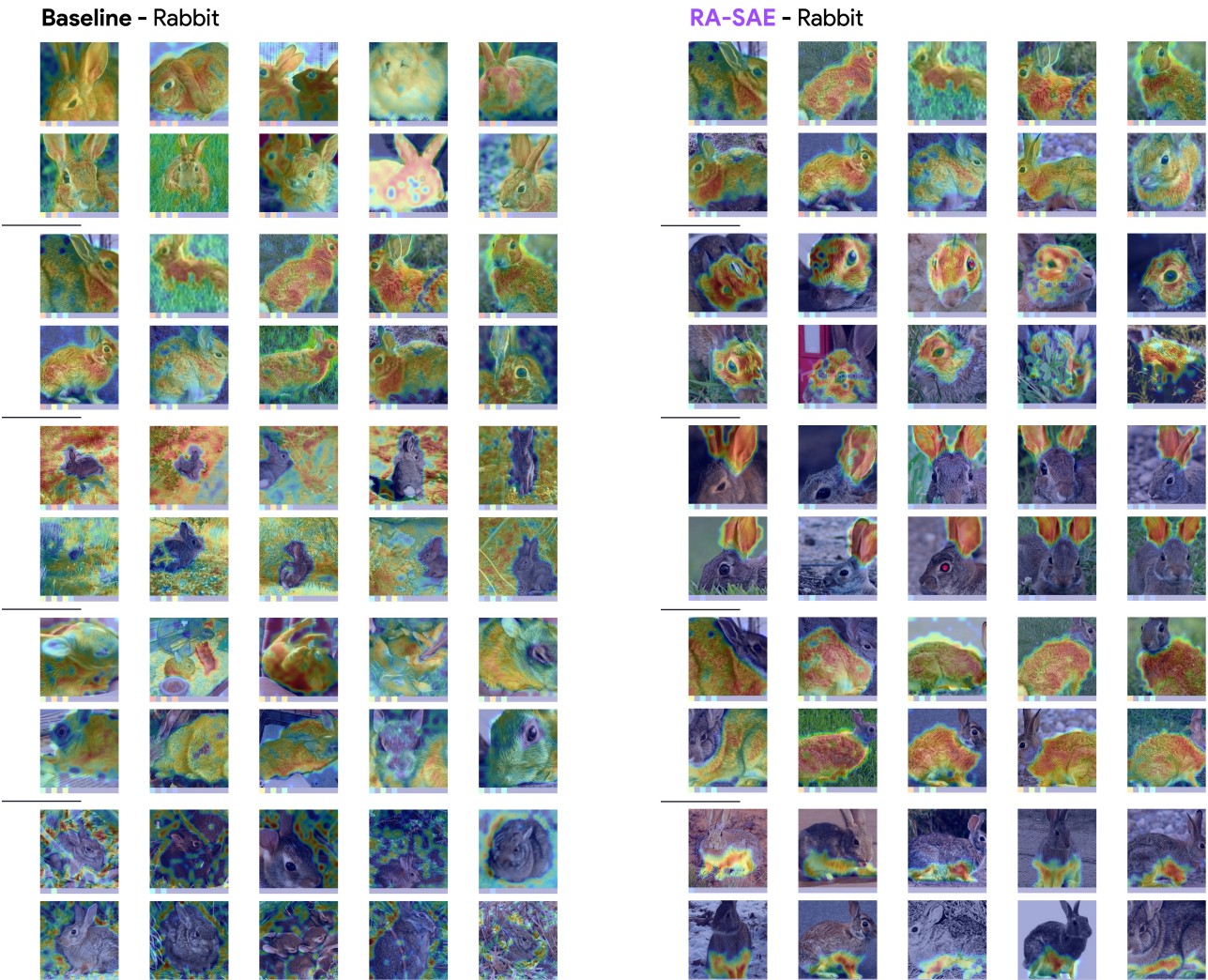

*Figure 9.* **Top-5 Concepts for the Rabbit Class in DinoV2.** The RA-SAE on top of TopK identifies distinct and fine-grained concepts, including rabbit ears, body, face, and paws. These concepts exhibit greater structure and granularity compared to those found by the unconstrained TopK SAE (baseline) method.

• A cluster of concepts capturing complex hand positions, ranging from hands in pockets to hands resting on another person (e.g., on a shoulder). • An abstract "under" concept cluster, linking entities such as birds, zebras, felines, and airplanes, while highlighting the lower regions of objects. • A cluster representing fine-grained facial concepts in animals, including ears, eyebrows, and cheeks.

In addition, we identified surprising and specific concepts among the $16,000$ dictionary atoms, shown in Figure **??**. These include: **A)** A concept highlighting tokens corresponding to shadows of dogs, suggesting that DinoV2 may use shadows as a feature, potentially contributing to its depth estimation or 3D reasoning capabilities. **B)** A "barber" concept that activates exclusively on tokens representing barbers but not on the individuals receiving haircuts or shaves. **C)** A fine-grained visual concept that activates along the contours or edges of flower petals.

Finally, we present the top-5 concepts associated with the rabbit class on DinoV2 in Figure 9. The concepts learned by RA-SAE are distinct and exhibit greater structure compared to their TopK counterparts. For instance, RA-SAE successfully identifies separate concepts for rabbit ears, body, face, and paws, demonstrating its ability to disentangle fine-grained features within a class. These results suggest that RA-SAE provides a more organized and meaningful decomposition of concepts.

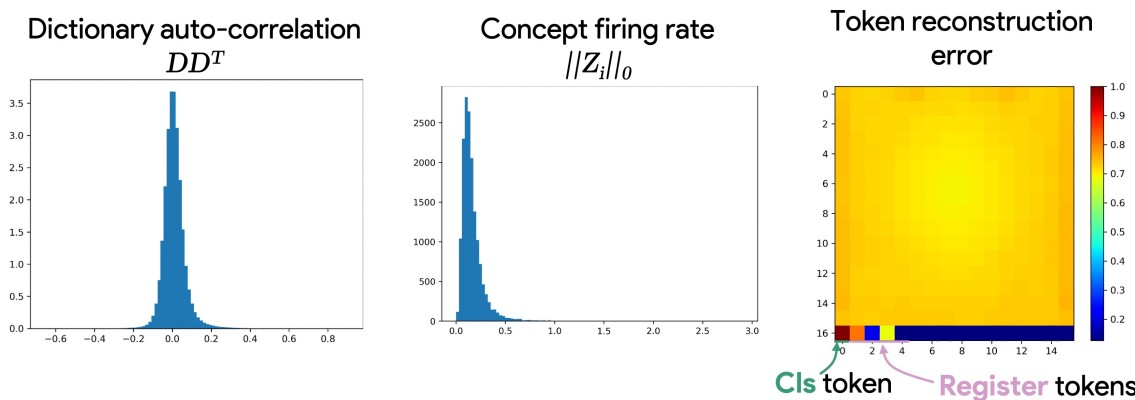

*Figure 10.* **Performance, Auto-Correlation, and Firing Rate of DinoV2 for Archetypal SAE.** The Archetypal SAE analyzed here demonstrates a well-structured dictionary with low auto-correlation, indicating that the dictionary atoms are not collinear. The firing rate exhibits a long-tail distribution, and reconstruction error is uniformly distributed across tokens, except for the CLS token, which shows the highest reconstruction error.

## C. Extended Related Work on Archetypal Analysis

Archetypal analysis (AA) was introduced by Cutler and Breiman in 1994 as a method to represent observations as convex combinations of extremal "pure types" called archetypes (Cutler & Breiman, 1994). Each archetype lies on the boundary of the data's convex hull and is itself constrained to be a convex combination of data points, yielding an interpretable factorization where data are explained in terms of extreme exemplars. This approach was proposed as an alternative to principal component analysis for uncovering latent structure, providing data-like representative factors instead of orthogonal directions (Cutler & Stone, 1997). Shortly after its introduction, AA was extended to spatio-temporal settings by identifying archetypes that vary over time (Stone & Cutler, 1996) and (Prabhakaran et al., 2012) proposed an automatic model selection criterion for AA, alleviating the need to fix $k$ a priori. After its initial formulation, AA did not gain widespread popularity, partly due to computational limitations and the availability of more established techniques like $k$-means or NMF. Nonetheless, several works in the 2000s applied AA successfully in scientific domains. For instance (Chan et al., 2003) applied AA to astronomical spectra. In genetics, (Gimbernat-Mayol et al., 2022) found that archetypes learned from human genotype data corresponded to ancestral population prototypes. AA was also utilized in socio-economic and bibliometric analyses (Seiler & Wohlrabe, 2013), and to model player behavior in video games (Sifa & Bauckhage, 2013). Various enhancements to the basic AA model have been proposed, (Eugster & Leisch, 2011) introduced weighted and robust AA, (Seth & Eugster, 2016) proposed to treats archetypes as latent factors in a probabilistic generative model. In a similar vein, contemporary work has developed AA for binary data specifically (Wedenborg & Mørup, 2025) and (Epifanio et al., 2019) tackled the presence of missing values. Likewise (Moliner & Epifanio, 2019) formulated robust AA for multivariate functional data.

Another important direction has been the development of non-linear and kernel variants of AA (Mørup & Hansen, 2012; Bauckhage & Manshaei, 2014). These kernel and multi-layer approaches connect AA with manifold learning techniques. (Javadi & Montanari, 2019) cast NMF as a special case of AA under separability assumptions. In term of algorithmic and scalability improvements (Chen et al., 2014) proposed a fast active-set algorithm, implemented in the SPAMS toolbox, making AA tractable for large-scale computer vision tasks. (Bauckhage et al.) adapted a Frank–Wolfe algorithm for AA, reducing runtime. (Mair & Brefeld, 2019) proposed coreset approximations to speed up archetype discovery, and (Abrol & Sharma, 2020) developed a greedy AA algorithm that extended naturally to robust and kernelized variants. Finally, (Mei et al., 2018) proposed an online algorithm for streaming data scenarios.

Mairal et al.'s task-driven dictionary learning framework inspired supervised AA models (Mairal et al., 2011). Related topic modeling approaches such as SurvLDA (Dawson & Kendziorski, 2012) and anchor-based models (Ding et al., 2016; Arora et al., 2013) similarly identify extremal structure. More recently, deep learning has been integrated with AA to handle complex, non-linear data. Wynen et al. (Wynen et al., 2018) applied AA to deep features for archetypal style analysis in artwork. Bauckhage et al. (Bauckhage et al.) described AA as an autoencoder. Keller et al. (Keller et al., 2020) developed a VAE-based deep AA that accommodates supervision. Applications are widespread. In vision, AA has been used to discover prototypical images (Thurau & Bauckhage, 2009), representative behaviors (Fotiadou et al., 2017), and fMRI brain patterns (Hinrich et al., 2016). In hyperspectral imaging, recent contributions such as SUnAA (Rasti et al., 2023) and entropic descent AA (Zouaoui et al., 2023) set new benchmarks. In NLP, AA was used in multi-document summarization (Canhasi & Kononenko, 2016) and anchor-based topic modeling (Ding et al., 2016; Arora et al., 2013).

**Comparison to Our Work**    Our method departs from AA by constraining **only the decoder** to the convex hull of data, while using a linear encoder, maintaining full compatibility with modern SAEs architectures like TopK and JumpReLU. Unlike AA, which jointly optimizes over convex atoms and codes, we retain end to end SAE training. Our goal is not AA approximation or generation, but improved stability and semantic consistency of SAEs. We further introduce a scalable relaxation of the convex constraint, adapted to large-scale training and distinct from prior AA relaxations (e.g., (Mørup & Hansen, 2012)).

# D. Formal Definitions of Metrics

To comprehensively evaluate SAEs and their archetypal variants, we have defined in Sec.5.1 a set of metrics that assess four key dimensions of the SAEs: (*i*) sparse reconstruction, (*ii*) consistency, (*iii*) structure in the dictionary ($D$), and (*iv*) structure in the codes ($Z$).

## D.1. Sparse Reconstruction

As explained in the main paper, these metrics evaluate the ability of the model to accurately reconstruct activations while enforcing sparsity constraints. We believe they are well understood and already used by the interpretability community. The **Reconstruction Error** ($R^2$) measures the fidelity of the reconstruction by quantifying how well the learned dictionary approximates the input activations:

$$R^2 = 1 - \frac{||\boldsymbol{A} - \hat{\boldsymbol{A}}||_F^2}{||\boldsymbol{A} - \bar{\boldsymbol{A}}||_F^2}, \quad (5)$$

with $\hat{\boldsymbol{A}} = \boldsymbol{ZD}$ the predicted activation and $\bar{\boldsymbol{A}}$ the mean activation matrix. Essentially, $R^2$ measures how much we improve on explaining variance upon the best possible predictor that uses only a single bias. The **Dead Codes** measure the fraction of dictionary atoms that remain unused across the dataset, highlighting inefficiencies in the learned representation, for a set of $n$ codes and $k$ concepts $\boldsymbol{Z} \in \mathbb{R}^{n \times k}$:

$$\text{Dead Codes} = 1 - \frac{1}{k}|| \sum_{i}^{n} \boldsymbol{Z}_i ||_0 \quad (6)$$

## D.2. Consistency

The second category of metrics assesses how consistent and well-grounded the learned dictionary is. Specifically, we evaluate (**i**) the **Stability** of the learned solution across training runs and (**ii**) its proximity to real data, measured via the **Out-of-Distribution (OOD) Score**.

**Stability**, introduced in Eq. 2, quantifies how consistent the learned dictionary remains when training is repeated with different initializations. Given two independently trained dictionaries, $\boldsymbol{D}$ and $\boldsymbol{D}'$, stability is defined as:

$$\text{Stability}(\boldsymbol{D}, \boldsymbol{D}') = \max_{\boldsymbol{\Pi} \in \mathcal{P}(n)} \frac{1}{n} \text{Tr}(\boldsymbol{D}^\mathsf{T} \boldsymbol{\Pi} \boldsymbol{D}'), \quad (7)$$

where $\mathcal{P}(n)$ is the set of $n \times n$ signed permutation matrices. A score of $1$ indicates perfect alignment—each concept in $\boldsymbol{D}$ has a direct equivalent in $\boldsymbol{D}'$, while a score of $0$ implies that all concepts are seed-specific and change arbitrarily across runs.

A looser measure of stability is the **Max Cosine Similarity**, which only considers the best-matching concept between two training runs:

$$\text{Max Cosine} = \max_{i,j}\langle \boldsymbol{D}_i, \boldsymbol{D}'_j \rangle. \quad (8)$$

This metric provides an upper bound on alignment but does not enforce global consistency across the dictionary.

Beyond stability, we assess how well the learned dictionary aligns with real data. The **Out-of-Distribution (OOD) Score** measures the deviation of dictionary atoms from real data points by computing the cosine similarity between each dictionary atom $\boldsymbol{D}_i$ and its closest real activation $\boldsymbol{A}_j$:

$$\text{OOD Score} = 1 - \frac{1}{k} \sum_{i=1}^{k} \max_{j \in [n]}\langle \boldsymbol{D}_i, \boldsymbol{A}_j \rangle. \quad (9)$$

A score of $0$ indicates that every dictionary atom $\boldsymbol{D}_i$ exactly matches an existing data point $\boldsymbol{A}_j$, meaning the model purely reconstructs real activations. Notably, methods like Separable-NMF (Gillis, 2020), which enforce the "pure-pixel" assumption by explicitly selecting dictionary atoms from the dataset, naturally achieve an OOD score of $0$.

Together, these metrics provide a comprehensive evaluation of how stable, interpretable, and grounded the learned dictionary remains across training runs and relative to real data.

## D.3. Structure in the Dictionary ($D$)

The third category assesses the internal organization of the dictionary, providing insights into its effective dimensionality, redundancy, and (we hope) overall interpretability. Unlike reconstruction or consistency metrics, which evaluate external properties of the learned dictionary, these metrics focus on how well-formed and structured the set of concepts is. Again, a dictionary with lower effective dimensionality suggests structure like compositionality and/or hierarchical concepts. **Stable Rank** is the first metric we propose that provides an effective measure of the intrinsic dimensionality of the dictionary:

$$\text{Stable Rank} = \frac{||\boldsymbol{D}||_F^2}{||\boldsymbol{D}||_2^2}. \quad (10)$$

Unlike the traditional matrix rank, which is sensitive to numerical precision (all dictionaries are nearly full rank in practice because of numerical error), the stable rank remains well-behaved even in high-dimensional settings, serving as a smooth proxy for rank estimation. **Effective Rank** offers an alternative perspective by measuring the entropy of the singular value distribution of $\boldsymbol{D}$:

$$\text{Eff. Rank} = \exp\left( -\sum_{i=1}^{k} \sigma_i \log \sigma_i \right), \quad (11)$$

where $\sigma_i$ are the **normalized** singular values of $D$ (i.e., $\sum_i \sigma_i = 1$). An effective rank close to $k$ suggests that all dictionary atoms are equally important, whereas a low effective rank implies that only a few dominant concepts capture most of the variation in the data. A fully orthogonal dictionary (thus not overcomplete) would achieve an effective rank of $k$. **Coherence** is a common notion in dictionary learning and compress sensing, it quantifies redundancy between dictionary atoms by measuring the maximum pairwise cosine similarity:

$$\text{Coherence} = \max_{i \neq j} |D_i^\top D_j|. \quad (12)$$

We still admit that each $D_i$ is on the $\ell_2$ ball. Lower coherence indicates that dictionary atoms are more diverse and span independent directions, which is desirable for disentangled representations. Conversely, high coherence suggests that multiple dictionary atoms encode nearly identical features, reducing the efficiency of the learned basis. Notably, coherence is closely related to the concept of *mutual incoherence* in compressed sensing (Donoho, 2006), where low-coherence bases are preferred for sparse signal recovery.

### D.4. (*iv*) Structure in the Codes ($Z$)

The last category measures how much structure we have in the codes. While dictionary structure (***iii***) focuses on the learned basis $D$, the structure of the codes $Z$ determines how these dictionary atoms are used to reconstruct activations. A well-structured encoding should exhibit meaningful combinations of concepts while avoiding redundancy and destructive interference. **Connectivity** measures the diversity of concept usage by counting the number of *unique co-activations* within the code matrix:

$$\text{Connectivity} = 1 - (\frac{1}{d^2}||Z^\top Z||_0). \quad (13)$$

This metric quantifies how many distinct pairs of concepts $(i, j)$ are activated together across samples. A high connectivity score suggests that a broad range of concepts can be meaningfully combined, leading to more complex representations. Conversely, low connectivity implies a highly structured representation, and in some sense a group-sparsity representation where only a subset of concepts can fire together. We note that connectivity in sparse coding has been linked to compositionality (Olshausen & Field, 1996). We believe that none of the SAEs currently studied achieve interesting performance on this metric. Finally, **Negative Interference** quantifies the extent to which co-activated concepts cancel each other out, reducing their effectiveness:

$$\text{Neg. Inter.} = ||\text{ReLU}(-(Z^\top Z) \odot (DD^\top))||_2. \quad (14)$$

Where $\odot$ is the Hadamard product, this metric captures cases where two concepts $i$ and $j$ frequently co-activate (as measured by $Z^\top Z$), yet their dictionary atoms are negatively correlated (indicated by a negative dot product in $DD^\top$). The ReLU function ensures that only destructive interactions are counted, where activation of both concepts leads to mutual cancellation rather than constructive combination. A high negative interference score suggests that the learned dictionary contains redundant or antagonistic concepts. In extreme cases, we could imagine a strong negative interference can lead to concept pairs that consistently suppress each other to comply with some sparsity constraint.

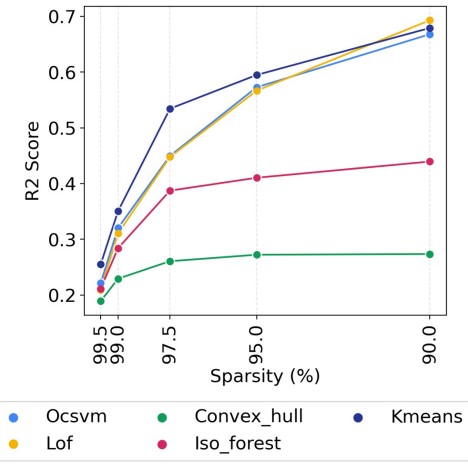

*Figure 11.* **Comparison of Distillation Methods.** To scale up the Archetypal SAE, it is impractical to utilize the entire data matrix $\mathbf{A}$ for identifying archetypes. Instead, we first reduce the dataset to a smaller subset of points, denoted as $\mathbf{C}$, and construct the archetypes/dictionary elements from this reduced set. Among the distillation methods evaluated, K-Means proves to be the most effective approach, generating points within the convex hull of the data and achieving strong performance scores. These experiments were conducted on DINOv2 using an Archetypal SAE without relaxation.

## E. Distilling $A$ into $C$

As a recall, Archetypal SAEs construct dictionary atoms as convex combinations of data points,

$$\boldsymbol{D} = \boldsymbol{W}\boldsymbol{A}, \quad \text{with} \quad \boldsymbol{W} \in \Omega_{k,n}, \quad \boldsymbol{A} \in \mathbb{R}^{n \times d}, \quad (15)$$

requiring access to the full data matrix $\boldsymbol{A}$. However, the original problem is intractable, as the number of points $n$ requires storing and processing millions of activations at each gradient step, which is computationally prohibitive, particularly for large-scale datasets of tokens. To address this, we proposed in Sec. 4 a *distillation* step, reducing $\boldsymbol{A}$ to a compact subset $\boldsymbol{C}$. The dictionary $\boldsymbol{D}$ is then formed using only $\boldsymbol{C} \in \mathbb{R}^{n' \times d}$, with $n' \ll n$, ensuring a tractable optimization while remaining representative of the original distribution.

We investigated five different methods to distill $\boldsymbol{A}$ into $\boldsymbol{C}$, as illustrated in Fig. 11:

- **K-Means**: Groups data into $m$ clusters and selects centroids as representatives. This ensures that $\boldsymbol{C}$ remains within the convex hull of $\boldsymbol{A}$ while capturing its most frequent patterns.
- **Local Outlier Factor (LOF)** (Breunig et al., 2000): Identifies statistically atypical points compared to their neighbors, helping remove rare or extreme cases.
- **Isolation Forest (Iso)** (Liu et al., 2008): Uses recursive

partitioning to isolate anomalies, providing an efficient method for detecting outliers.
- **Convex Hull on Reduced Dimensions**: Computes the convex hull of $\boldsymbol{A}$ after projecting it onto its 10 principal components via PCA. This ensures that extreme points defining the overall shape of the distribution are retained while reducing computational complexity.
- **One-Class SVM (OC-SVM)** (Scholkopf et al., 1999): A support vector method that learns a boundary around high-density regions, effectively isolating representative points while filtering outliers.

Among these, **K-Means emerges as the most effective method**. Furthermore, while convex hull approaches theoretically guarantee coverage of extreme points, their computational cost scales poorly with high dimensions, making them impractical for large datasets.

## F. Implementation Details

We provide an efficient PyTorch implementation of the relaxed dictionary learning module used in Archetypal SAEs. Given a distilled set of centroids $\boldsymbol{C} \in \mathbb{R}^{n' \times d}$, our goal is to construct a dictionary $\boldsymbol{D}$ as a convex combination of elements in $\boldsymbol{C}$, while allowing a controlled degree of relaxation via additive perturbations.

- The weight matrix $\boldsymbol{W} \in \mathbb{R}^{k \times n'}$ is constrained to the probability simplex, ensuring convex combinations of centroids. This is enforced via a `convex_param` function that projects $\boldsymbol{W}$ onto the simplex using ReLU and row-wise normalization.
- The relaxation term $\boldsymbol{\delta} \in \mathbb{R}^{k \times d}$ allows mild deviations from strict convexity. To prevent excessive drift, $\boldsymbol{\delta}$ is regularized by adaptively scaling its norm to remain within a relaxation factor.
- Dictionary atoms are computed as $\boldsymbol{D} = \boldsymbol{W}\boldsymbol{C} + \boldsymbol{\delta}$.
- The model supports gradient updates for $\boldsymbol{W}$ and $\boldsymbol{\delta}$ while keeping $\boldsymbol{C}$ fixed.

```python
class ArchetypalDictionary(nn.Module):
  """Relaxed Archetypal SAE (RA-SAE) dictionary.

  Constructs a dictionary where each atom is a convex combination of data
  points from C, with a small relaxation term Λ constrained by δ.
  """

  def __init__(self, C, k, delta=1.0):
    """
    Parameters
    ----------
    C : Tensor
        Candidate archetypes of shape (n', d).
    k : int
        Number of dictionary atoms.
    delta : float
        Upper bound on the norm of the relaxation term Λ.
    """
    super().__init__()
    n_prime, d = C.shape
    self.register_buffer("C", C) # store C as a fixed buffer (non-trainable)
    self.W = nn.Parameter(torch.eye(k, n_prime)) # trainable param (row-stochastic)
    self.Lambda = nn.Parameter(torch.zeros(k, d)) # small relaxation term
    self.delta = delta # constraint on the relaxation term norm

  def forward(self, Z):
    """
    Parameters
    ----------
    Z : Tensor
        Sparse codes of shape (n, k).

    Returns
    -------
    Tensor
        Reconstructed activations of shape (n, d).
    """
    with torch.no_grad():
      # ensure W remains row-stochastic (positive and row sum to one)
      W = torch.relu(self.W)
      W /= W.sum(dim=-1, keepdim=True)
      self.W.data = W

      # enforce the norm constraint on Λ to limit deviation from conv(C)
      norm_Lambda = self.Lambda.norm(dim=-1, keepdim=True) # norm per row
      scaling_factor = torch.clamp(self.delta / norm_Lambda, max=1) # safe scaling factor
      self.Lambda *= scaling_factor # scale Λ to satisfy ||Λ|| ≤ δ

    # compute the dictionary as a convex combination plus relaxation
    D = self.W @ self.C + self.Lambda

    return Z @ D
```

*Figure 12.* **Detailed Pytorch code for Relaxexd Archetypal SAE (RA-SAE).**

# G. Theoretical properties of Archetypal Dictionary

In this section, we provide theoretical insights into Archetypal Sparse Autoencoders (A-SAEs) by addressing three key aspects: (*i*) why standard SAEs can produce dictionaries that drift away from the data manifold, (*ii*) a geometric interpretation of the A-SAE solution and the conditions under which distillation is optimal, and (*iii*) bounds on the stability, rank, and out-of-distribution (OOD) score of A-SAEs.

A simple explanation for why standard SAEs may drift away from the data manifold can be found by examining the gradient descent (GD) update rule for the dictionary $D$. Given a dataset $A \in \mathbb{R}^{n \times d}$, nonnegative codes $Z \geq 0 \in \mathbb{R}^{n \times k}$, and a dictionary $D \in \mathbb{R}^{k \times d}$ with unit-norm rows ($\|D_j\|_2 = 1$), the standard SAE optimization problem is:

$$\min_{Z, D} \|A - ZD^{\mathsf{T}}\|_F^2 \quad \text{s.t.} \quad Z \geq 0, \quad \|D_j\|_2 = 1. \quad (16)$$

The gradient of the reconstruction loss with respect to $D$ is given by:

$$\nabla_D \|A - ZD^{\mathsf{T}}\|_F^2 = 2(Z^{\mathsf{T}}ZD - Z^{\mathsf{T}}A). \quad (17)$$

This gradient can be decomposed into two components:

$$\Delta D = \underbrace{Z^{\mathsf{T}}A}_{\text{data-anchored term}} - \underbrace{Z^{\mathsf{T}}ZD}_{\text{out-of-data term}}.$$

The first term, $Z^{\mathsf{T}}A$, pulls dictionary atoms toward a conic combination of data points, anchoring them to $\mathrm{cone}(A)$. However, the second term, $Z^{\mathsf{T}}ZD$, introduces a drift effect that pushes the dictionary away from the data, as it depends on the correlations within $Z$ and the original seed $D$.

Empirically, in high-dimensional settings where the codes $Z$ exhibit sufficient variability could induce the second term to dominate. This could explain why classical SAEs have a relatively low OOD score, as shown in Sec. 1 and why minor perturbations in initialization or the training set can potentially yield different dictionaries, as the second term is entirely dependent of the seed.

## G.1. Geometric interpretation of A-SAE

**Proposition G.1** (Archetypal Dictionary, Convex and Conic Hulls)**.** *Given $A \in \mathbb{R}^{n \times d}$ as a set of $n$ data points and $W \in \Omega_{k,n}$ as any row-stochastic matrix, parameterizing $D = WA$ ensures that each concept $D_i$ lies within the convex hull of the data, i.e., $D_i \in \mathrm{conv}(A)$ for all $i \in [k]$. Moreover, for any nonnegative codes $Z \geq 0$, the reconstruction $ZD$ lies within the conic hull of the data, i.e., $ZD \subseteq \mathrm{cone}(A)$. More generally, let $C \in \mathbb{R}^{n' \times d}$ such that $\mathrm{conv}(C) \subseteq \mathrm{conv}(A)$ be a subset of $A$ or any set of points within $\mathrm{conv}(A)$. Then $D' = WC$ satisfies*

*$D'_i \in \mathrm{conv}(C) \subseteq \mathrm{conv}(A)$ and $ZD' \subseteq \mathrm{cone}(A)$. Finally, if $C$ includes the extreme points of $A$, then no representational power is lost.*

*Proof.* Since $W$ is row-stochastic, each $D_i = W_i A$ is a convex combination of the rows of $A$, i.e., $D_i \in \mathrm{conv}(A)$. Furthermore, for nonnegative $Z$, we have $ZD = (ZW)A$, with $ZW \geq 0$, which implies that $ZD \subseteq \mathrm{cone}(A)$. With $D' = WC$ for $\mathrm{conv}(C) \subseteq \mathrm{conv}(A)$, each row $D'_i$ lies within $\mathrm{conv}(C) \subseteq \mathrm{conv}(A)$. Lastly, if $C$ contains the extreme points (Boyd & Vandenberghe, 2004) of $A$, then by simple application of the Krein-Milman theorem (Lax, 2014), $\mathrm{conv}(C) = \mathrm{conv}(A)$ and $\mathrm{cone}(ZWC) = \mathrm{cone}(A)$, ensuring no loss in expressivity. $\square$

The constraints imposed by **A-SAE** on $D$ lead to a straightforward yet crucial geometric property: each dictionary atom $D_i$ remains within the convex hull of the data. This result, while simple, has interesting implications for some of the metrics we study, notably stability and OOD. In fact, we will now see that this implies a bounded OOD score, prevents the rank of the dictionary (and thus its structure) to higher than the rank of the data, and induces some loose form of algorithmic stability.

## G.2. Stability of Archetypal Dictionary

**Proposition G.2** (Geometric Stability of Archetypal Dictionaries)**.** *Let $A, A' \in \mathbb{R}^{n \times d}$ be two data matrices such that $\|A - A'\|_F \leq \varepsilon$. Suppose $W, W' \in \Omega_{k,n}$ are row-stochastic matrices (i.e. each row belongs to the probability simplex). Define the archetypal dictionaries $D = WA$ and $D' = W'A'$. Then,*

$$\|D - D'\|_F \leq \sqrt{k}\,\varepsilon + 2\sqrt{k}\,\min\left(\|A\|_F, \|A'\|_F\right).$$

*Proof.* Using triangle inequality, we have

$$\|D - D'\|_F = \|WA - W'A'\|_F \leq \|WA - WA'\|_F + \|WA' - W'A'\|_F.$$

We bound each term separately, since $D$ and $D'$ are archetypal dictionaries, $W$ is row-stochastic, hence $\|W\|_2 \leq \sqrt{k}$. Therefore,

$$\|WA - WA'\|_F = \|W(A - A')\|_F \leq \|W\|_2\,\|A - A'\|_F,$$

and $\|W\|_2\,\|A - A'\|_F \leq \sqrt{k}\,\varepsilon$. For the second term,

$$\|WA' - W'A'\|_F = \|(W - W')A'\|_F \leq \|W - W'\|_F\,\|A'\|_F,$$

with $||\boldsymbol{W} - \boldsymbol{W}'||_F \le 2\sqrt{k}$ summing these yield:

$$||\boldsymbol{D} - \boldsymbol{D}'||_F \le \sqrt{k}\,\varepsilon + 2\sqrt{k}\,||\boldsymbol{A}'||_F.$$

It is straightforward to repeat the above process with a $||\boldsymbol{A}'||_F$ factor in the right hand side. Taking the minimum over the two factors then completes the proof. $\square$

The key observation is that a row-stochastic matrix $\boldsymbol{W}$ cannot stretch data arbitrarily, imposing a natural control on how $\boldsymbol{D} = \boldsymbol{W}\boldsymbol{A}$ changes if $\boldsymbol{A}$ is slightly perturbed. By contrast, in the unconstrained setting, where the dictionary $\boldsymbol{D}$ is free to move anywhere (subject only to norms or regularization), there is no comparably simple bound ensuring stability. Even small perturbations in $\boldsymbol{A}$ (or in the random seed, initialization, etc.) can shift the solution significantly: as there is no requirement that $\boldsymbol{D}$ stay close to $\mathrm{conv}(\boldsymbol{A})$, the learned atoms can drift to entirely different regions of the space.

### G.3. Controlled Rank and Stability in A-SAE Dictionaries

We now show that the rank of the dictionary obtained using the Archetypal parametrization are inherently controlled. Specifically, the rank of the dictionary cannot exceed the rank of the data. This property is interesting as it prevents the dictionary from becoming arbitrarily complex, promoting solutions that are more structured and aligned with the data. Consequently, the dictionary is more likely to uncover meta-concepts or low-rank representations.

**Proposition G.3** (Rank Bound of Archetypal Dictionaries). *Let $\boldsymbol{A} \in \mathbb{R}^{n \times d}$ be a data matrix with $\mathrm{rank}(\boldsymbol{A}) = r \le d$, assuming $n \gg d$ and $k \gg d$. Let $\boldsymbol{W} \in \Omega_{k,n}$ be a row-stochastic matrix, and define the dictionary as $\boldsymbol{D} = \boldsymbol{W}\boldsymbol{A}$. Then, the rank of $\boldsymbol{D}$ is bounded by the rank of the data:*

$$\mathrm{rank}(\boldsymbol{D}) \le \min(\mathrm{rank}(\boldsymbol{A}), \mathrm{rank}(\boldsymbol{W})) \le d.$$

*Proof.* Since $\boldsymbol{D} = \boldsymbol{W}\boldsymbol{A}$, the column space of $\boldsymbol{D}$ is contained in the column space of $\boldsymbol{A}$, implying

$$\mathrm{rank}(\boldsymbol{D}) \le \mathrm{rank}(\boldsymbol{A}) = r.$$

Additionally, since $\boldsymbol{W} \in \mathbb{R}^{k \times n}$ is row-stochastic, its rank is at most $\min(k, n)$, giving

$$\mathrm{rank}(\boldsymbol{W}) \le \min(k, n).$$

it follows that

$$\mathrm{rank}(\boldsymbol{D}) = \mathrm{rank}(\boldsymbol{W}\boldsymbol{A}) \le \min(r, \mathrm{rank}(\boldsymbol{W})).$$

$\square$

### G.4. Bounding OOD score with Archetypal Constraints

We now demonstrate that the OOD measure of dictionary atoms obtained under some assumption is inherently lower-bounded. Specifically, the measure is directly tied to the weights in the row-stochastic matrix, with a maximum value of 1 achieved when a dictionary atom perfectly aligns with a data point. This property is particularly interesting as it ensures that dictionary atoms remain well-grounded in the data. Furthermore, the sparsity of the weight matrix plays a crucial role in maintaining orthogonality, thereby preserving the plausibility of the assumption and the robustness of the bounds.

**Proposition G.4** (OOD Measure with Non-Interfering Archetypes). *Let $\boldsymbol{A} \in \mathbb{R}^{n \times d}$ be our data matrix where each point $\boldsymbol{A}_i$ is normalized ($||\boldsymbol{A}_i||_2 = 1$ for all $i \in [n]$). Let our archetypal dictionary $\boldsymbol{D} = \boldsymbol{W}\boldsymbol{A}$, where $\boldsymbol{W} \in \mathbb{R}^{k \times n}$ is a row-stochastic matrix ($\boldsymbol{W} \in \Omega_{k,n}$). We assume non-interfering Archetypes, meaning two non-orthogonal archetypes cannot be active at the same time (but can exist in the bank of points $\boldsymbol{A}$). Formally, for each row $\boldsymbol{W}_i$, the active rows of $\boldsymbol{A}$ (those $\boldsymbol{A}_j$ with $\boldsymbol{W}_{ij} > 0$) are orthogonal, i.e.,*

$$\langle \boldsymbol{A}_j, \boldsymbol{A}_{j'} \rangle = 0 \quad for\ j \ne j'\ and\ \boldsymbol{W}_{ij}, \boldsymbol{W}_{ij'} > 0.$$

*Then, the out-of-distribution (OOD) measure for each $\boldsymbol{D}_i$ admits the upper bound:*

$$\mathrm{OOD}(\boldsymbol{D}_i) \le 1 - \max_{j \in [n]} \boldsymbol{W}_{ij}.$$

*Proof.* By definition, $\boldsymbol{D}_i = \sum_{j=1}^{n} \boldsymbol{W}_{ij}\boldsymbol{A}_j$, so

$$\langle \boldsymbol{D}_i, \boldsymbol{A}_j \rangle = \left\langle \sum_{k=1}^{n} \boldsymbol{W}_{ik}\boldsymbol{A}_k, \boldsymbol{A}_j \right\rangle = \sum_{k=1}^{n} \boldsymbol{W}_{ik}\langle \boldsymbol{A}_k, \boldsymbol{A}_j \rangle.$$

Under the orthogonality assumption $\langle \boldsymbol{A}_k, \boldsymbol{A}_j \rangle = 0$ for $k \ne j$, only the $k = j$ term remains:

$$\langle \boldsymbol{D}_i, \boldsymbol{A}_j \rangle = \boldsymbol{W}_{ij}\langle \boldsymbol{A}_j, \boldsymbol{A}_j \rangle.$$

Since $||\boldsymbol{A}_j||_2 = 1$, we have $\langle \boldsymbol{A}_j, \boldsymbol{A}_j \rangle = 1$, so:

$$\langle \boldsymbol{D}_i, \boldsymbol{A}_j \rangle = \boldsymbol{W}_{ij}.$$

By definition:

$$||\boldsymbol{D}_i||_2^2 = \left\| \sum_{j=1}^{n} \boldsymbol{W}_{ij}\boldsymbol{A}_j \right\|_2^2.$$

By the orthogonality of the active rows of $\boldsymbol{A}$, the contribu-

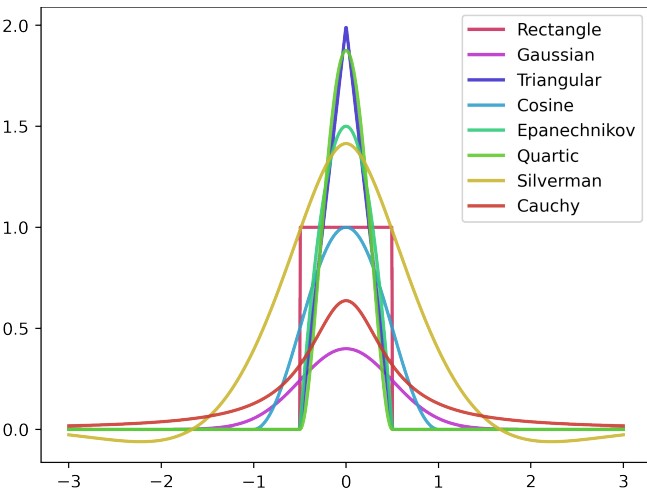

*Figure 13.* Example of kernel functions for JumpReLU over the interval $[-3, 3]$ with a bandwidth of 1. The Silverman kernel produced more stable and positive results, leading to its selection for all experiments with a smaller bandwidth of $10^{-2}$.

tions of different rows do not interfere, so:

$$\|\boldsymbol{D}_i\|_2^2 = \sum_{j=1}^{n} \boldsymbol{W}_{ij}^2 \|\boldsymbol{A}_j\|_2^2.$$

Since $\|\boldsymbol{A}_j\|_2 = 1$, this simplifies to:

$$\|\boldsymbol{D}_i\|_2^2 = \sum_{j=1}^{n} \boldsymbol{W}_{ij}^2.$$

Substituting these yields our bound:

$$\mathrm{OOD}(\boldsymbol{D}_i) = 1 - \max_{j\in[n]} \frac{\langle \boldsymbol{D}_i, \boldsymbol{A}_j \rangle}{\|\boldsymbol{D}_i\|_2} \leq 1 - \max_{j\in[n]} \boldsymbol{W}_{ij}.$$

$\square$

As a notable special case, we observe that $\mathrm{OOD}(\boldsymbol{D}_i) = 0$ when $\boldsymbol{W}_{ij} = 1$ for some $j \in [n]$, as the dictionary atom aligns perfectly with a data point. In practice, the sparsity of $\boldsymbol{W}$ plays a crucial role: it limits interference between (non-orthogonal) components, ensuring that the orthogonality assumption remains plausible and that the derived bounds hold robustly.

## H. Kernel for JumpReLU

JumpReLU (Rajamanoharan et al., 2024) is a recently introduced activation mechanism for SAEs designed to optimize $\ell_0$ sparsity by controlling the discontinuities of ReLU through a parameter $\theta$. Its optimization relies on a kernel for density estimation. To assess the effect of kernel choice,

we evaluated several options, including Gaussian, Cauchy, and Silverman, on DinoV2. As shown in Figure 13, the Silverman kernel consistently yielded the most stable and accurate reconstruction results. For our experiments, we selected the Silverman kernel with a bandwidth of $10^{-2}$, although the choice of kernel appears to have only a modest impact on performance.

## I. Soft Identifiability Benchmark

In this appendix, we provide additional details regarding the experimental setup and evaluation criteria used in the Soft Identifiability Benchmark. We recall that the goal is to assess the ability of SAEs to recover distinct concepts from synthetic image mixtures, where the underlying generative factors are known.

We generate twelve synthetic datasets using Midjourney API[4]. For each of these datasets, we programmatically create 4,000 images. These images are constructed by collaging four distinct objects selected from the predefined set, such as different types of gems. Each dataset is generated from between 9 and 20 unique objects, with the dictionary size set exactly to the number of true generative factors (unique object number). An example of some datasets used in the benchmark is shown in Figure 14.

Each dataset is split into a training set of 2,000 images and a test set of 2,000 images. The images are processed through a pre-trained vision model, in our case DinoV2, ResNet50, SigLIP and ViT. The resulting pooled activations serve as the input representations for the SAE.

**Metrics.** To quantitatively evaluate identifiability, we define an accuracy metric that measures whether each object class in the dataset is correctly assigned a distinct concept in the SAE dictionary. Given an image, we pass it through the vision model and then through the trained SAE to obtain a concept-label pair $(\boldsymbol{z}, \boldsymbol{y})$, where $\boldsymbol{z} \in \mathbb{R}^k$ represents the $k$ learned concept activations, and $\boldsymbol{y} \in \mathbb{R}^c$ denotes the $c$ ground-truth class labels.

We define the accuracy for class $j$ as:

$$\mathrm{Accuracy}_j = \max_{\lambda\in\mathbb{R}, i\in[k]} \mathbb{P}_{(\boldsymbol{z},\boldsymbol{y})}((z_i > \lambda) = y_j), \quad (18)$$

where $\lambda$ is a threshold determining whether a concept is activated.

To find an appropriate $\lambda$, we use the empirical percentiles of the concept activations $\boldsymbol{Z}$, ranging from the 1st to the 100th percentile. This ensures that the threshold is adaptive to the distribution of activations, optimizing for the best classification accuracy.

---

[4]www.midjourney.com

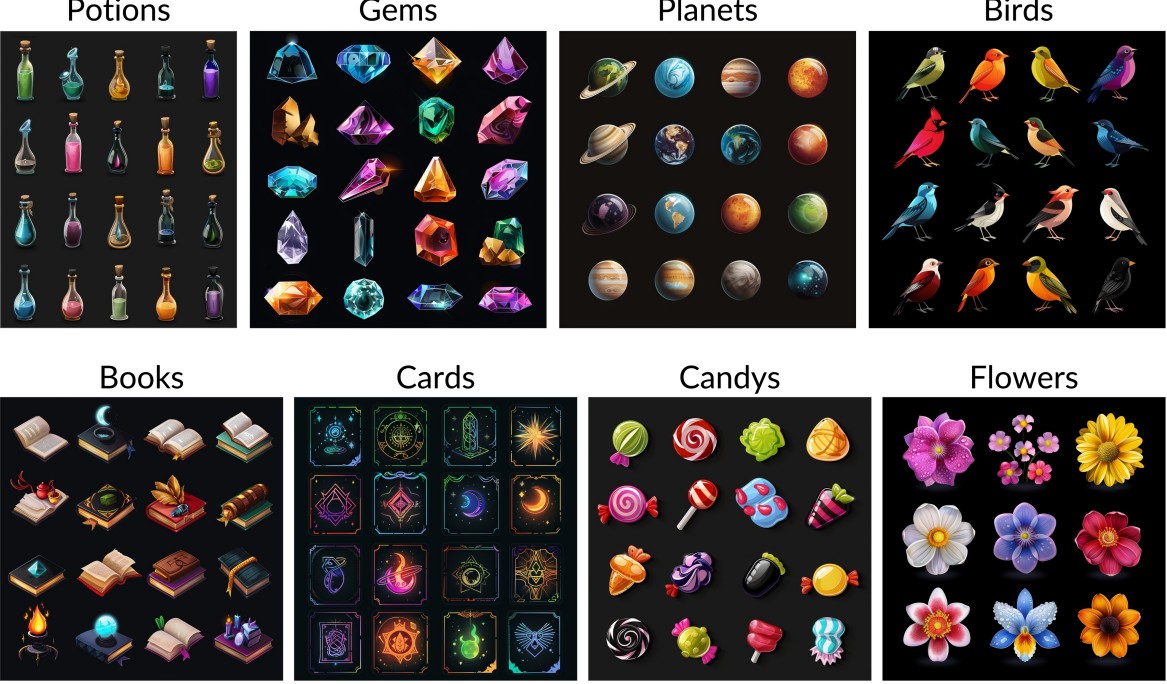

Potions  Gems  Planets  Birds

Books  Cards  Candys  Flowers

*Figure 14.* Examples of synthetic datasets used in the Identifiability Benchmark.

## I.1. Complete Results

The full set of results across all methods and datasets is provided in Table 4. We also provide a comprehensive breakdown, including per-dataset accuracy scores and additional analysis.

*Table 4.* Accuracy Scores for Various Methods Across Models and Classes

| Model | Method | Animals | Birds | Books | Candy | Cards | Cocktails | Flowers | Gems | Landscapes | Planets | Potions | Watches | Avg |
|---|---|---|---|---|---|---|---|---|---|---|---|---|---|---|
| **DINO** | KMeans | 0.7679 | 0.7678 | 0.7715 | 0.7709 | 0.7724 | 0.8095 | 0.6670 | 0.8137 | 0.7147 | 0.7786 | 0.8104 | 0.7693 | 0.7678 |
| | ICA | 0.8113 | 0.7967 | 0.8099 | 0.8182 | 0.8212 | 0.8296 | 0.7497 | 0.8569 | 0.7807 | 0.8002 | 0.8297 | 0.8068 | 0.8092 |
| | Sparse PCA | 0.8033 | 0.7919 | 0.8013 | 0.7939 | 0.8129 | 0.8245 | 0.7126 | 0.8322 | 0.7733 | 0.8022 | 0.8307 | 0.7986 | 0.7981 |
| | SVD | 0.8037 | 0.7916 | 0.8018 | 0.7935 | 0.8142 | 0.8245 | 0.7116 | 0.8320 | 0.7716 | 0.8023 | 0.8301 | 0.7978 | 0.7979 |
| | SemiNMF | 0.8175 | 0.8059 | 0.8560 | 0.8660 | 0.8464 | 0.8516 | 0.7360 | 0.8564 | 0.8111 | 0.8261 | 0.8569 | 0.8264 | 0.8297 |
| | ConvexNMF | 0.7726 | 0.7658 | 0.7739 | 0.7759 | 0.7711 | 0.8108 | 0.6264 | 0.8163 | 0.7013 | 0.7688 | 0.8205 | 0.7711 | 0.7645 |
| | PCA | 0.8037 | 0.7916 | 0.8018 | 0.7935 | 0.8142 | 0.8245 | 0.7116 | 0.8320 | 0.7716 | 0.8023 | 0.8301 | 0.7978 | 0.7979 |
| | Vanilla | 0.7968 | 0.7878 | 0.8087 | 0.8161 | 0.8062 | 0.8468 | 0.7202 | 0.8518 | 0.7601 | 0.8262 | 0.8385 | 0.7977 | 0.8047 |
| | TopK | 0.7906 | 0.7942 | 0.8104 | 0.8283 | 0.8243 | 0.8407 | 0.7728 | 0.8501 | 0.7744 | 0.8184 | 0.8387 | 0.8191 | 0.8135 |
| | Jump | 0.7863 | 0.7956 | 0.7970 | 0.8174 | 0.8121 | 0.8251 | 0.7389 | 0.8418 | 0.7504 | 0.7989 | 0.8438 | 0.8042 | 0.8010 |
| | A-SAE | 0.9433 | 0.9413 | 0.9692 | 0.9722 | 0.9750 | 0.8901 | 0.9590 | 0.9590 | 0.9677 | 0.9277 | 0.9129 | 0.9606 | **0.9482** |
| | RA-SAE | 0.9402 | 0.9313 | 0.9703 | 0.9614 | 0.9686 | 0.8905 | 0.9613 | 0.9503 | 0.9666 | 0.9227 | 0.9094 | 0.9642 | 0.9447 |
| **ResNet** | PCA | 0.8249 | 0.8086 | 0.8556 | 0.8643 | 0.8622 | 0.8501 | 0.7908 | 0.8446 | 0.7814 | 0.8203 | 0.8326 | 0.8134 | 0.8291 |
| | KMeans | 0.7670 | 0.7647 | 0.7720 | 0.7668 | 0.7669 | 0.8123 | 0.6418 | 0.8109 | 0.6964 | 0.7713 | 0.8109 | 0.7686 | 0.7624 |
| | ICA | 0.8169 | 0.8231 | 0.8315 | 0.8358 | 0.8295 | 0.8852 | 0.7902 | 0.8900 | 0.8378 | 0.8206 | 0.8619 | 0.8213 | 0.8370 |
| | Sparse PCA | 0.8256 | 0.8162 | 0.8599 | 0.8689 | 0.8661 | 0.8501 | 0.7937 | 0.8461 | 0.7841 | 0.8230 | 0.8330 | 0.8155 | 0.8318 |
| | SVD | 0.8249 | 0.8086 | 0.8556 | 0.8643 | 0.8623 | 0.8501 | 0.7910 | 0.8445 | 0.7816 | 0.8201 | 0.8324 | 0.8134 | 0.8291 |
| | SemiNMF | 0.8259 | 0.8274 | 0.8536 | 0.8611 | 0.8419 | 0.8577 | 0.7307 | 0.8632 | 0.8198 | 0.8195 | 0.8486 | 0.8432 | 0.8327 |
| | ConvexNMF | 0.7647 | 0.7648 | 0.7699 | 0.7669 | 0.7659 | 0.8086 | 0.6102 | 0.8111 | 0.6939 | 0.7686 | 0.8098 | 0.7644 | 0.7582 |
| | Vanilla | 0.8062 | 0.8143 | 0.8270 | 0.8228 | 0.8397 | 0.8412 | 0.7380 | 0.8393 | 0.7894 | 0.8147 | 0.8475 | 0.8198 | 0.8167 |
| | TopK | 0.8078 | 0.8083 | 0.8384 | 0.8269 | 0.8176 | 0.8496 | 0.7437 | 0.8397 | 0.7753 | 0.8193 | 0.8482 | 0.8052 | 0.8150 |
| | Jump | 0.7949 | 0.8027 | 0.8082 | 0.8274 | 0.8144 | 0.8345 | 0.6648 | 0.8373 | 0.7485 | 0.8161 | 0.8287 | 0.8084 | 0.7988 |
| | A-SAE | 0.9633 | 0.9703 | 0.9638 | 0.9673 | 0.9713 | 0.9738 | 0.9894 | 0.9722 | 0.9658 | 0.9342 | 0.9539 | 0.9315 | **0.9631** |
| | RA-SAE | 0.9613 | 0.9577 | 0.9694 | 0.9834 | 0.9709 | 0.9625 | 0.9640 | 0.9497 | 0.9629 | 0.9371 | 0.9554 | 0.9479 | 0.9602 |
| **SigLIP** | PCA | 0.8253 | 0.7957 | 0.8264 | 0.8030 | 0.8157 | 0.8286 | 0.7367 | 0.8270 | 0.7678 | 0.7931 | 0.8291 | 0.8261 | 0.8062 |
| | KMeans | 0.7733 | 0.7672 | 0.7733 | 0.7691 | 0.7742 | 0.8094 | 0.6690 | 0.8121 | 0.7171 | 0.7733 | 0.8101 | 0.7724 | 0.7684 |
| | ICA | 0.8372 | 0.8171 | 0.8151 | 0.8254 | 0.8382 | 0.8341 | 0.7918 | 0.8333 | 0.8420 | 0.8056 | 0.8362 | 0.8158 | 0.8243 |
| | Sparse PCA | 0.8251 | 0.7962 | 0.8374 | 0.8021 | 0.8152 | 0.8269 | 0.7369 | 0.8286 | 0.7676 | 0.7941 | 0.8286 | 0.8240 | 0.8069 |
| | SVD | 0.8253 | 0.7957 | 0.8264 | 0.8030 | 0.8157 | 0.8286 | 0.7367 | 0.8269 | 0.7678 | 0.7931 | 0.8291 | 0.8261 | 0.8062 |
| | SemiNMF | 0.8574 | 0.8386 | 0.8413 | 0.8559 | 0.8521 | 0.8393 | 0.7780 | 0.8524 | 0.7979 | 0.8320 | 0.8389 | 0.8463 | 0.8358 |
| | ConvexNMF | 0.7706 | 0.7706 | 0.7769 | 0.7704 | 0.7687 | 0.8090 | 0.6314 | 0.8121 | 0.6972 | 0.7718 | 0.8134 | 0.7745 | 0.7639 |
| | Vanilla | 0.8240 | 0.8160 | 0.8097 | 0.8202 | 0.8429 | 0.8436 | 0.7167 | 0.8493 | 0.7775 | 0.7976 | 0.8433 | 0.8108 | 0.8126 |
| | TopK | 0.8254 | 0.8190 | 0.8215 | 0.8439 | 0.8664 | 0.8460 | 0.7576 | 0.8547 | 0.8022 | 0.8253 | 0.8483 | 0.8364 | 0.8289 |
| | Jump | 0.8199 | 0.8351 | 0.8048 | 0.8206 | 0.8222 | 0.8367 | 0.7307 | 0.8493 | 0.7813 | 0.7946 | 0.8352 | 0.8269 | 0.8131 |
| | A-SAE | 0.9727 | 0.9613 | 0.9517 | 0.9686 | 0.9753 | 0.9445 | 0.9622 | 0.9669 | 0.9553 | 0.9479 | 0.9457 | 0.9704 | **0.9602** |
| | RA-SAE | 0.9655 | 0.9654 | 0.9411 | 0.9681 | 0.9749 | 0.9366 | 0.9632 | 0.9594 | 0.9543 | 0.9325 | 0.9546 | 0.9861 | 0.9585 |
| **ViT** | PCA | 0.7994 | 0.8107 | 0.8226 | 0.8258 | 0.7963 | 0.8352 | 0.7780 | 0.8274 | 0.7556 | 0.8029 | 0.8199 | 0.8164 | 0.8075 |
| | KMeans | 0.7706 | 0.7719 | 0.7744 | 0.7776 | 0.7752 | 0.8095 | 0.6721 | 0.8134 | 0.7120 | 0.7841 | 0.8099 | 0.7722 | 0.7702 |
| | ICA | 0.8134 | 0.8251 | 0.8243 | 0.8437 | 0.8159 | 0.8623 | 0.7983 | 0.8539 | 0.7998 | 0.8229 | 0.8415 | 0.8188 | 0.8267 |
| | Sparse PCA | 0.8003 | 0.8109 | 0.8225 | 0.8287 | 0.7963 | 0.8342 | 0.7780 | 0.8282 | 0.7563 | 0.8022 | 0.8201 | 0.8208 | 0.8082 |
| | SVD | 0.7994 | 0.8107 | 0.8226 | 0.8258 | 0.7963 | 0.8351 | 0.7780 | 0.8274 | 0.7556 | 0.8029 | 0.8201 | 0.8164 | 0.8075 |
| | SemiNMF | 0.8232 | 0.8414 | 0.8436 | 0.8410 | 0.8598 | 0.8661 | 0.8133 | 0.8584 | 0.8064 | 0.8418 | 0.8492 | 0.8629 | 0.8423 |
| | ConvexNMF | 0.7709 | 0.7685 | 0.7684 | 0.7714 | 0.7689 | 0.8117 | 0.6254 | 0.8132 | 0.7017 | 0.7736 | 0.8134 | 0.7738 | 0.7634 |
| | Vanilla | 0.8169 | 0.8548 | 0.8459 | 0.8234 | 0.8438 | 0.8481 | 0.7312 | 0.8598 | 0.7512 | 0.8249 | 0.8285 | 0.8387 | 0.8223 |
| | TopK | 0.8494 | 0.8430 | 0.8577 | 0.8264 | 0.8405 | 0.8503 | 0.7849 | 0.8534 | 0.7910 | 0.8215 | 0.8310 | 0.8451 | 0.8328 |
| | Jump | 0.8042 | 0.7995 | 0.8316 | 0.8156 | 0.8064 | 0.8377 | 0.6872 | 0.8515 | 0.7681 | 0.8143 | 0.8300 | 0.8172 | 0.8053 |
| | A-SAE | 0.9647 | 0.9847 | 0.9519 | 0.9838 | 0.9733 | 0.9578 | 0.9938 | 0.9488 | 0.9384 | 0.9369 | 0.9325 | 0.9719 | **0.9615** |
| | RA-SAE | 0.9699 | 0.9847 | 0.9683 | 0.9696 | 0.9694 | 0.9620 | 0.9576 | 0.9455 | 0.9281 | 0.9484 | 0.9253 | 0.9745 | 0.9586 |

