# OpenReview forum: "Archetypal SAE: Adaptive and Stable Dictionary Learning for Concept Extraction in Large Vision Models"
_ICML.cc/2025/Conference — ICML 2025 poster_

### Official Review · Reviewer_DiCd · 2025-02-19

**Overall Recommendation:** 3

**Summary:**

The paper points out the issue of feature instability in SAE training and designs the A-SAE and RA-SAE methods. These methods restrict the SAE feature Z using an archetypal dictionary D, resulting in SAEs with more stable features. The paper also introduces metrics for evaluating SAEs: (i) sparse reconstruction, (ii) plausibility, (iii) structure in the dictionary (D), and (iv) structure in the codes (Z). In summary, RA-SAE matches classical SAEs in reconstruction while significantly improving stability, structure, and alignment with real data.

## update after rebuttal

Thanks for your rebuttal. I'd like to keep my score as my concern in Experimental Designs Or Analyses is still remains. What I care most is whether this SAE can work properly in LLMs and VLMs, which takes the stage of Interpretability research for their complexity and unexplanation. So I thank it's lack of novelty and practical significance that only applies current experiment settings.

**Claims And Evidence:**

Yes.

**Essential References Not Discussed:**

No.

**Experimental Designs Or Analyses:**

Yes. The experiment used five models (DINOv2, SigLip, ViT, ConvNeXt, ResNet50), but most of these models are based on convolutional neural networks (CNN) or self-supervised learning methods. It is unclear whether they represent all types of models. For instance, models with completely different architectures (such as Transformers) may yield different results. I would expect the paper to include supplementary experiments on smaller transformer models, such as Gemma-2B or Llava-7B, even if these experiments are conducted in a pure text modality. I will consider **raising the score** based on this point.

**Methods And Evaluation Criteria:**

Yes.

**Other Comments Or Suggestions:**

In sections 4. Towards Archetypal SAEs and 5. Experiments, the paper does not clearly describe the specific **training process** of the model. I would expect the inclusion of pseudocode or diagrams to help readers better understand the training process. If this aspect were improved, I would consider raising the score.

**Other Strengths And Weaknesses:**

Strength: This paper propose a new-type SAE with stable feature dictionary and provide corresponding math proof, which is a point with novelty.

Weaknesses: A logic problem of this paper is that it doesn't do some interpretability experiment which showing the strength of the Archetypal SAE. Yes I find some visualizing result and case study in appendix, but I propose that more experiment in different dataset and model will be better. Also, this paper is lack of ablation study. If this aspect were improved, I would consider raising the score.

**Questions For Authors:**

No.

**Relation To Broader Scientific Literature:**

The main idea of the paper and the benchmark method is related to the proposal and evaluation of TopK SAE in "Scaling and evaluating sparse autoencoders."

**Theoretical Claims:**

Yes, no obvious theoretical issues were found in the paper.

---

> ### Author Rebuttal · Authors · 2025-04-01
>
> Thank you for the feedback! We are glad you found the stability contributions and mathematical formulation of Archetypal SAEs to be novel and well-motivated. Below we address specific comments.
>
> ---
>
> > **On Diversity of Analyzed Pretrained Models**
>
> We agree that evaluating across a range of model types is important! However, we emphasize that our analyzed models already cover both **Transformer architectures** (DINOv2, ViT, SigLIP) and **convolutional ones** (ConvNext and ResNet50). Further, we highlight SigLIP is a **vision–language model**, i.e., it is trained with text supervision. Thus, our experiments *already* include Transformer models and multi-modal pretraining pipelines. We promise to highlight this diversity of evaluations in the final version of the paper.
>
> ---
>
> > **On Relation to TopK SAE and Prior Work**
>
> We appreciate your reference to prior work such as "Scaling and Evaluating Sparse Autoencoders." However, we would like to clarify that our **contributions are orthogonal** to the specific choice of SAE variant. Archetypal SAEs are **a general framework** that can be applied on top of *any* SAE architecture, including Vanilla, TopK, JumpReLU, or BatchTopK. That is, we are not proposing another variant of TopK SAEs, nor analyzing TopK specifically. Instead, we introduce a geometric anchoring method that complements existing sparsity-inducing architectures by improving stability and structure. We will revise the paper to clarify this distinction more directly.
>
> ---
>
> > **On Qualitative Interpretability Results**
>
> We fully agree with this comment: qualitative interpretability is critical! In response to this point and similar comments from other reviewers, we have **moved several qualitative results from the appendix into the main paper**, including concept clusters (Fig. 10), fine-grained decompositions of ImageNet classes (Fig. 8), and exotic concepts in DINOv2 (Fig. 7). These changes will be reflected in the next paper update, where we plan to *use the final page of the camera-ready version* to showcase such qualitative examples across models and datasets, highlighting the types of structured concepts discovered by RA-SAE. We thank you for encouraging more emphasis on this front.
>
> ---
>
> > **On Pseudocode**
>
> Thank you for this helpful suggestion! We have now included a **detailed pseudocode block in Appendix D** showing the full training procedure for RA-SAE, including convex constraint handling, relaxation updates, and sparse reconstruction. We have also added a minimalistic version of the pseudocode to the main paper. These changes will be reflected in the next paper update.
>
> ---
>
> > **On Ablation Studies**
>
> Thank you for the comment! Unfortunately, we could not understand precisely which ablations the reviewer intended to suggest for us to analyze—if the reviewer can clarify this point, we are happy to add relevant experiments to the final paper. For now, we emphasize that our method introduces **only one novel hyperparameter**—the relaxation parameter δ (lambda). We already provide extensive ablations for the same in Fig. 4 and Table 2 (plausibility benchmark). Additionally, we also provide experiments titrating following settings:
> - Different **dictionary sizes** (k from 512 to 32K),
> - Multiple **vision architectures** (5 total, with both uni- and multi-modal pretraining),
> - Multiple **SAE baselines** (Vanilla, TopK, JumpReLU),
> - Different **distillation strategies** for the convex anchor set (Appendix C.1).
>
> Please let us know if there is a specific titration / ablation experiment you believe would help strengthen our work—we would be happy to add it to the final paper.
>
> ---
> ---
>
> **Summary:** Thank you again for the constructive feedback. We hope our responses help address your raised questions, and our changes to the paper, e.g., the inclusion of pseudocode and qualitative results in the main paper, merit score increase inline with your comments! Please let us know if you have any further questions.

---

### Official Review · Reviewer_1Jmo · 2025-03-03

**Overall Recommendation:** 3

**Summary:**

The authors find that current SAE architectures (ReLU, Jump-ReLU, TopK) exhibit instability: the learned concepts differ between runs, even on the same data. They measure this with a new metric:
$$\text{max}_{\Pi} \frac{1}{n} \text{Tr}(D^\intercal \Pi D')$$

Where $\Pi$ is the optimal alignment between $D$ and $D'$. Compared to other dictionary learning methods, SAEs are significantly less stable, but achieve much better sparsity and reconstruction error. To mitigate this instability, the authors propose Archetypal SAEs (and eventually Relaxed Archetypal SAEs) which constrains the decoder matrix $W_\text{dec}$ such that each row of $W_\text{dec}$ is a convex combination of the rows of $A$ (the original activation set). A-SAEs and RA-SAEs achieve significantly improved stability, and RA-SAEs achieve similar reconstruction error and sparsity metrics as current SAE architectures.

The authors validate their findings on a variety of vision transformers and CNNs on ImageNet, then introduce a synthetic benchmark to identify if SAEs (both existing and their proposed A-SAE and RA-SAE) can reliably identify core features.

**Claims And Evidence:**

The authors claim that instability is a problem to be solved. Intuitively, I agree that stability between training runs is a desired property. Different random seeds should not lead to wildly different outcomes. However, I am not convinced that the proposed stability metric is the best way to measure this. At their core, SAEs are a method to interpret large pre-trained neural networks. While reconstruction error and sparsity are two metrics, nearly all prior work admits that these metrics are mere proxies for the true goal of "interpretability." Similarly, I feel that the proposed stability metric does not really reflect the downstream goals of SAEs: to interpret neural networks.

Can you provide a convincing argument that the instability in existing SAE architectures prevents us from reliably interpreting neural networks?


Second, could larger dictionaries solve instability?
As the size of the dictionary approaches infinity, if the dictionary avoids duplicating rows, then stability will eventually be solved.
What about simply as $k$ goes from 1K to 32K?
How does stability change?

**Essential References Not Discussed:**

N/A. Concurrent work in this area of SAEs for vision models might be nice to cite but not necessary.

* Sparse Autoencoders for Scientifically Rigorous Interpretation of Vision Models (https://arxiv.org/abs/2502.06755)
* Universal Sparse Autoencoders: Interpretable Cross-Model Concept Alignment (https://arxiv.org/abs/2502.03714)

**Experimental Designs Or Analyses:**

I don't understand the motivation behind the plausibility benchmark (Section 5.2).
Why would we want SAEs to recover true classification directions?
Furthermore, why would we use $k=512$ when the models have more than 512 dimensions AND ImageNet-1K has 1000 classes?
Again, we use SAEs to interpret concepts internal to the network, not to measure their task-specific utility (classification).
There is no reason for an SAE with $k=32K$ to choose concepts aligned with the linear classifier.
The decomposed features could *compose* into a concept needed by the linear classifier.

I am unconvinced that the soft identifiability benchmark (Section 5.3) makes sense.
It feels very arbitrary: sythetic images (OOD for the ViTs), a dictionary that has a fixed size that's equal to the true number of underlying concepts and a tuned threshold $\lambda$ for classification.
The scores are also not very convincing: RA-SAE achieves 0.94 for DINOv2 and Vanilla achieves 0.80.
What is a good score?
What is a bad score?

I would be unable to reproduce the experiments based solely on the main text and appendix, but the design set out in the main text is reasonable.
The only concern I have is $k=5$.
Prior work in SAEs for LLMs often uses much larger expansions.

I have the following questions:

* How do you choose the tradeoff term $\lambda$ for balancing between reconstruction error and sparsity for vanilla SAEs?
* What other hyperparamters for training do you use (learning rate, LR warmup, activation normalization, etc)?
* What layer of the vision models do you record?

**Methods And Evaluation Criteria:**

ViTs, CNNs, and ImageNet all make sense as an evaluation benchmark.
In fact, extending to CNNs is very interesting and I would be interested in the details.
It would of course be better to have another dataset (iNat2021?) to validate that these methods don't apply only to ImageNet but it's sufficient.

In terms of evaluation criteria, I am unsure the OOD score in C.2 is well motivated.
Why would I want my dictionary atoms to match real activations?
Intuitively, what's important is that the dictionary atoms can be reliably *composed* to match real activations, not that each atom actually matches an activation.

**Other Comments Or Suggestions:**

The spacing between table captions and text is too small.
Please add some additional space back to make the paper more readable.

**Other Strengths And Weaknesses:**

The related work is an excellent list of papers about sparse coding and traditional approaches to extracting concepts from distributions of activations.

**Questions For Authors:**

No questions.

**Relation To Broader Scientific Literature:**

This paper identifies a core problem in dictionary learning (stability) and proposes a new method, taking inspiration from prior work where necessary. I am satisfied with the presentation of related scientific literature.

**Theoretical Claims:**

I unfortunately am unable to check the theoretical claims in Appendix F.

---

> ### Author Rebuttal · Authors · 2025-04-01
>
> Thank you for the detailed and thoughtful review! We appreciate your engagement. Below, we address specific comments.
>
> ---
>
> > **Stability, why It matters, and how we measure it**
>
> We agree that interpretability—not reconstruction or sparsity per se—is the end goal of SAEs. However, we argue stability is *necessary* for interpretability to be meaningful: if concepts identified across SAE training runs vary drastically (as we show in Fig. 1, 3), then, since the basis along which explanations are developed (i.e., the concepts) changes, our ''explanations'' of how a model produces its outputs will change drastically as well. This raises the question whether we are identifying the ''right'' explanations in the first place.
>
> To this end, we emphasize that we see our paper as a first step towards identifying and addressing the broader challenge of instability in SAE training: grounded in classical dictionary learning literature, we offer a reasonable metric to quantify how dissimilar concepts identified via SAEs across training runs are, and propose a plausible solution to the problem (archetypal SAEs). However, we do not intend to suggest our defined metric for measuring instability is optimal—we merely argue that, to the extent linearly encoded concepts are identified by SAEs, measuring average cosine similarity after optimal matching is a reasonable notion of instability. We are certain useful metrics will be defined by future work, and we will make sure to emphasize this as a possible future direction in the final version of the paper.
>
> ---
>
> > **On the OOD Metric (Appendix C.2)**
>
> Thank you for the feedback! We will better clarify the motivation for the OOD metric in the final version of the paper. In brief, our argument is that if learned concepts lie entirely in the nullspace of real activations, they cannot possibly affect downstream layers. Thus, concept directions should maintain nontrivial similarity with data activations.
>
> ---
>
> > **On the Plausibility Benchmark (Section 5.2)**
>
> We emphasize our motivation for proposing the plausibility benchmark was to evaluate SAEs’ ability to identify interpretable features with a natural dataset—since it is hard to define ground truth for such an evaluation, we proposed the use of class label as a reasonable ground-truth to evaluate how interpretable concepts identified via SAEs are. However, we do *not* argue a score of 1.0 should be achieved on this evaluation—SAEs should not reduce to mere linear classifiers. Our argument is merely that given that class-labels of a dataset can be deemed as coarse concepts, we should expect at least some dictionary atoms to (partially) model such class hyperplanes. We will clarify this motivation in the final version of the paper.
>
> On reporting of results with a dictionary of size 512 on this benchmark: we note our goal here was to evaluate how results are affected with scaling of the dictionary size, i.e., size 512 is just one of 7 configurations we report on the benchmark. We nevertheless agree that since ImageNet is 1K classes, other reported configurations in this table are more meaningful—we will hence remove the 512 one.
>
> ---
>
> > **On the Soft Identifiability Benchmark (Section 5.3)**
>
> This benchmark offers a synthetic analogue of our plausibility benchmark, where an SAE’s ability to model more fine-grained concepts is evaluated by defining a setting with known generative factors. Next to a full-fledged human study to study interpretability of identified concepts, we argue such a setting offers meaningful insight into different SAEs’ ability to identify interpretable features.
>
> Clarification about evaluation setup: The setup used in this benchmark (dictionary size = true factor count, threshold sweep for λ)  is designed to provide an upper bound on performance by offering a maximally generous setting for all SAEs—not an operational, deployment setting. Meanwhile, the evaluation score is defined as the percentage of generative factors represented by an SAE in its dictionary atoms.
>
> ---
>
> > **On Model Training Details and Hyperparameters**
>
> Thank you for catching this—detailed information about training protocol has now been added to Appendix D (summarized below).
>
> - SAEs’ training: 50 epochs with batch size 4096.
> - Learning rate: 5e-4 with cosine decay and linear warmup.
> - Activations’ location: penultimate layer, the layer used for downstream tasks (after final layer norm for DINOv2, SigLIP, ConvNeXt; no norm for ResNet50).
> - Vanilla SAE uses λ = 1.0 only when batch sparsity is higher from target else 0.0 (measured per-batch).
> - JumpReLU penalizes thresholds $\theta$ using a Silverman kernel.
>
> ---
>
> > **Suggested refs**
>
> Thank you—we have added citations to both!
>
> ---
>
> > **Qualitative Results**
>
> We are glad you enjoyed the qualitative examples in Appendix B! We promise to use the extra page offered for the camera-ready version to showcase additional qualitative clusters and concept visualizations.

---

> > ### Comment · Reviewer_1Jmo · 2025-04-03
> >
> > Agreed on your discussion of why stability matters. I think this is a nuanced, subtle point that is not perfectly articulated in the work. While I am sure that you have spent many hours working on how to communicate this point, I encourage you to further workshop the language or presentation to see if you can explain it in a way that is immediately obvious and intuitive to readers. On the other hand, perhaps I am the only reader who was not immediately convinced.
> >
> > > In brief, our argument is that if learned concepts lie entirely in the nullspace of real activations, they cannot possibly affect downstream layers
> >
> > I sort of agree with the intuition here: if concepts are orthogonal to *weight matrices*, then then cannot affect downstream activations. Are you arguing that if a particular concept vector was orthogonal to all real activations, then it must not interact with the weights at all? I like the overall idea but I am not clear on the math. I would greatly appreciate any insight you can provide here.
> >
> >
> >
> > > Our argument is merely that given that class-labels of a dataset can be deemed as coarse concepts, we should expect at least some dictionary atoms to (partially) model such class hyperplanes
> >
> > I still disagree. It's entirely possible to represent a class concept (vector) as a linear sum of other sub-class concepts (also vectors). I really don't think this is a good motivating example to demonstrate that (R)A-SAEs are better than vanilla SAEs.
> >
> >
> > > we argue such a setting offers meaningful insight into different SAEs’ ability to identify interpretable features.
> >
> > While I agree that it offers insight, I'm not convinced that this toy example is an accurate model of the real world of ViT/LLM activations.
> >
> > Thank you for the detailed reply and clarifications. I will leave my score at a 3.

---

> > > ### Author Response · Authors · 2025-04-06
> > >
> > > Thanks again for the continued engagement. It’s been clear you’ve really taken the time with this paper, and we’ve genuinely appreciated that. Concerning your remarks:
> > > Stability is not just a nice-to-have—it is a prerequisite for meaningful interpretation. The very notion of interpreting an ANN using a learned concept basis hinges on the identity of that basis being reliable. If repeated runs yield non-corresponding concept sets, the analyst cannot determine whether any claimed interpretability is intrinsic to the model or an artifact of optimization stochasticity. We are not merely introducing a new metric—we are demonstrating that a foundational assumption of interpretability (that the extracted features are meaningful because they reflect something consistent about the model) breaks down under current SAE training regimes.
> > >
> > > Regarding your question about nullspace/OOD: **you are exactly right. Our point is not that a concept orthogonal to the classifier’s weights is "bad" per se, but that a concept orthogonal to all real activations is operationally useles**. If a learned direction has zero inner product with every activation in the data, it can never influence downstream activations. This is not a matter of faithfulness or human semantics—it is a linear algebraic fact.
> > >
> > > We agree with your point that class hyperplanes can be composed from lower-level features—this is precisely why **we don't expect every SAE feature to align with class weights. But we do expect a model that claims to find high-level semantic directions to exhibit some alignment with those hyperplanes**. In practice, RA-SAE achieves substantially higher alignment scores across models (Tab. 2), indicating better capture of coarse-grained semantics without sacrificing compositionality. We’ll revise the plausibility benchmark framing to emphasize this more clearly.
> > >
> > > Finally, while our synthetic benchmark is not a full proxy for real-world ViT representations, it enables controlled study of identifiability—something not possible with in-the-wild datasets. It exposes when an SAE fails to separate distinct sources in the input signal, and serves as an analogue to disentanglement evaluation in the representation learning literature (Locatello et al. 2019; Higgins et al. 2017). In future work, we plan to develop human-grounded evaluations—but establishing synthetic identifiability is a necessary first step.
> > >
> > > We hope this clarifies the points you raised. Thanks again for the back-and-forth—it’s been genuinely valuable to interact over these ideas. We’re hopeful the strength of the results and framing justifies a second look at the score.

---

### Official Review · Reviewer_hoxe · 2025-03-06

**Overall Recommendation:** 4

**Summary:**

Sparse autoencoders (SAEs) are a promising unsupervised learning approach to find relevant and interpretable concepts of representations, e.g., for language or vision models. This paper argues that concepts extracted from SAEs are unstable when a fix model is trained multiple times on the same dataset or trained on similar datasets. The address this issue, the authors propose to constrain the dictionary atoms to reside on the boundary of the convex hull of data. This is the idea of archetypal analysis and the authors propose to combine archetypal analysis and SAEs yielding A-SAEs. In their paper, data actually refers to the embedding of image data produced by a vision model, e.g., DINOv2 or ViT. By having the dictionary atoms spread out yet close to data (archetypal analysis idea), the authors ague / demonstrate that their A-SAE
- yields sparse reconstructions which are on-par with SOTA,
- is plausible as it yields dictionary $D$ atoms are close to data,
- has dictionary $D$ atoms that have meaningful directions,
- maintain structure in the codes $Z$.
The authors conduct several experiments to verify their claims.

## update after rebuttal
Based on the author responses, the discussion, their clarifications, and their committment to improve the paper, I reconsidered my evaluation and raised my score.

**Claims And Evidence:**

- ~~The first contribution/claim *"We identify a critical limitation of current SAEs: dictionaries are unstable across different training runs, compromising their reliability."* (lines 90-92) does not seem to be evaluated properly since I do not find any information on repeated/different training runs (different initialization) anywhere in the paper. This can also be seen in the tables which do not mention any measures of centrality (mean, median) and dispersion (std). An exception is Table 3 which mentions an average.~~
- ~~Within their second contribution/claim (lines 93-95), they apply parts of archetypal analysis within the context of sparse autoencoders. Parts, because only the dictionary is constrained to be within the convex hull of data. In archetypal analysis, also the reconstruction is constrained to be within the convex hull of the dictionary which forces the dictionary to reside on the boundary of the convex hull of data.~~
- Moreover, the proposed relaxed version (line 95) seems to already exist. The proposed usage of a reduced subset is also not novel. (For both, see Essential References Not Discussed).
- ~~As for their fourth contribution/claim (lines 99-101), they construct a new dataset but it is unclear if it will be openly released.~~

**Essential References Not Discussed:**

- The related work mentions many works about sparse coding and dictionary learning but **actually surprisingly little about archetypal analysis**. I expected to see at least the **works that combine archetypal analysis and autoencoders (or more generally neural networks)**, i.e.,
	- Wynen, Daan, Cordelia Schmid, and Julien Mairal. "Unsupervised learning of artistic styles with archetypal style analysis." Advances in Neural Information Processing Systems 31 (2018).
	- van Dijk, David, Daniel B. Burkhardt, Matthew Amodio, Alexander Tong, Guy Wolf, and Smita Krishnaswamy. "Finding archetypal spaces using neural networks." In 2019 IEEE International Conference on Big Data (Big Data), pp. 2634-2643. IEEE, 2019.
	- Keller, Sebastian Mathias, Maxim Samarin, Mario Wieser, and Volker Roth. "Deep archetypal analysis." In German Conference on Pattern Recognition, pp. 171-185. Cham: Springer International Publishing, 2019.
	- Keller, Sebastian Mathias, Maxim Samarin, Fabricio Arend Torres, Mario Wieser, and Volker Roth. "Learning extremal representations with deep archetypal analysis." International Journal of Computer Vision 129 (2021): 805-820.
- Lines 263-267: Specifically for archetypal analysis, **this was also/already shown** (along a way on how to compute $C$) in
	- Mair, Sebastian, Ahcene Boubekki, and Ulf Brefeld. "Frame-based data factorizations." In International Conference on Machine Learning, pp. 2305-2313. PMLR, 2017.
- Line 301 and Appendix D: There is actually **quite some related work on using a reduced subset $C$** instead of the full $A$ matrix which seems to be missing or neglected:
	- Mørup, Morten, and Lars Kai Hansen. "Archetypal analysis for machine learning and data mining." Neurocomputing 80 (2012): 54-63.
	- Mair, Sebastian, Ahcene Boubekki, and Ulf Brefeld. "Frame-based data factorizations." In International Conference on Machine Learning, pp. 2305-2313. PMLR, 2017.
	- Damle, Anil, and Yuekai Sun. "A geometric approach to archetypal analysis and nonnegative matrix factorization." Technometrics 59, no. 3 (2017): 361-370.
	- Mair, Sebastian, and Ulf Brefeld. "Coresets for archetypal analysis." Advances in Neural Information Processing Systems 32 (2019).
	- Han, Ruijian, Braxton Osting, Dong Wang, and Yiming Xu. "Probabilistic methods for approximate archetypal analysis." Information and Inference: A Journal of the IMA 12, no. 1 (2023): 466-493.
	- Mair, Sebastian, and Jens Sjölund. "Archetypal Analysis++: Rethinking the Initialization Strategy." Transactions on Machine Learning Research (2024).
- Lines 313-319: **A relaxation of archetypes was already proposed** in
	- Mørup, Morten, and Lars Kai Hansen. "Archetypal analysis for machine learning and data mining." Neurocomputing 80 (2012): 54-63.
- Line 302: If $A$ is normalized and also considering the assumption for measuring instability in Equation (2), the following paper seems to be relevant:
	- Mei, Jieru, Chunyu Wang, and Wenjun Zeng. "Online dictionary learning for approximate archetypal analysis." In Proceedings of the European Conference on Computer Vision (ECCV), pp. 486-501. 2018.
- Lines 1047-1051: **Computing the convex hull on reduced dimensions is also not new**:
	- Thurau, Christian, Kristian Kersting, and Christian Bauckhage. "Convex non-negative matrix factorization in the wild." In 2009 Ninth IEEE International Conference on Data Mining, pp. 523-532. IEEE, 2009.
	- Bauckhage, Christian, and Christian Thurau. "Making archetypal analysis practical." In Joint Pattern Recognition Symposium, pp. 272-281. Berlin, Heidelberg: Springer Berlin Heidelberg, 2009.

**Experimental Designs Or Analyses:**

- I do not find any information on repeated/different training runs (different initialization) anywhere in the paper. This can also be seen in the tables which do not mention any measures of centrality (mean, median) and dispersion (std).
	- The exception is Figure 3.
- I wonder why archetypal analysis is not a baseline.
- The experimental setup appears to be sound otherwise.

**Methods And Evaluation Criteria:**

- The main idea is to constrain the dictionary $D$ to reside within the convex hull of the embedding $A$. For this, a similar idea to archetypal analysis is used. However, due to scalability issues, a smaller version $C$ instead of $A$ is used. The authors argue for $K$-Means (line 302) and argue it is the most effective method (lines 1057-1058).
	- $K$-Means will select **cluster centers that lie inside the convex hull of data**. Thus, **the convex hull of $C$ will be smaller in volume than the convex hull of $A$**. This cannot be wanted. The difference in volume should be as small as possible to achieve a good approximation of $conv(C)=conv(A)$ (see also line 266).
	- There are many neglected ways of computing $C$ in a better and faster way yielding larger convex hulls of $C$ than $K$-Means, for example:
		- Damle, Anil, and Yuekai Sun. "A geometric approach to archetypal analysis and nonnegative matrix factorization." Technometrics 59, no. 3 (2017): 361-370.
		- Mair, Sebastian, and Ulf Brefeld. "Coresets for archetypal analysis." Advances in Neural Information Processing Systems 32 (2019).
	- Technically, even the initialization method FurthestSum can also be used to construct $C$:
		- Mørup, Morten, and Lars Kai Hansen. "Archetypal analysis for machine learning and data mining." Neurocomputing 80 (2012): 54-63.
	- **I believe that due to the smaller convex hull of $C$ (which is due to $K$-Means), the proposed relaxed variant is needed. If $C$ is chosen in a better way, the relaxation might be superfluous.**

**Other Comments Or Suggestions:**

- Inconsistent capitalization. Why is archetypal analysis capitalized in line 22 but not in line 80? Why is neural network capitalized (line 41)?
- Line 136: Equation (1).
- Lines 157-160: Backpropagation is computing a gradient for gradient-based optimization and gradient-based optimization can usually also be phrased in terms of mini-batch updates. I do not see how this is an argument against traditional methods and in favor of SAEs.
- Lines 161-185: Neural networks are also learned in multiple steps. Again, I do not see the argument.
- Line 190: Pareto
- Lines 202-203: The line spacing seems to be off.
- Figure 2: All we see is that with non-linear approaches we get lower errors than with linear approaches across various sparsity levels. This has nothing to do with scalability. (Why is Loss capitalized?)
- Line 191: Rephrase to $D, D' \in \mathbb{R}^{n \times d}$.
- Line 191: Why is $D \in \mathbb{R}^{n \times d}$? Consider lines 127-138. If $A \in \mathbb{R}^{n \times d}$, then $Z \in \mathbb{R}^{n \times k}$ and $D \in \mathbb{R}^{d \times k}$.
- Lines 328,368: Tab. 1 vs Table 2
- Tables 1,2,3: Table captions go above tables. This way the caption can also be easier visually separated from the text, see lines 353-354 for an example.
- Lines 353-356: Equation (18) upper bounds the error, no?
- Figure 5: In 2., it appears visually that the SAE reconstructs $X$ but it reconstructs $A$!
- Line 437: Why capitalized?
- Please unify the style of the references: arXiv paper have different styles, sometimes conferences come with abbreviations, sometimes not, sometimes just the abbreviation is provided, line 658 is missing the conference, sometimes arXiv is stated although the paper is published, etc.
- Line 770: Section A is empty.
- Lines 793-796: What is $k$?
- Lines 1071-1080: Why are there dashes?

**Other Strengths And Weaknesses:**

- The paper is mostly very well written.
- However, the clarity of some parts should be improved. Here are some examples:
	- Figure 1: At this point, the reader is unfamiliar with the notation $v$. Moreover, it is unclear what the three images per run and model represent. Please improve the caption.
	- Lines 139-143: Why different $k$'s, i.e., $k$ and $K$? Please also state what $k$ refers to/means. Currently, it is only done implicitly but the readability of the paper can be improved by stating its meaning clearly.
	- I think SAEs should be better formally introduced.
	- Line 256: Is $A \in \mathbb{R}^{n \times d}$ the data matrix in input space or is it the result of the function $f$ (line 114), i.e., not the data matrix but rather the latent-representation matrix?
	- Line 265: Here, $D$ is $k \times d$ which is different from line 138. Why?
	- Line 273: The reconstructions used to be $ZD^T$ (line 137).
	- Line 315: The variable $\Lambda$ is undefined.
	- Line 302: I think the reader has a hard time understanding that the $A$ matrix is a result from the five evaluated model and that the SAE is trained on the embedding of $A$, especially when the reader has the standard autoencoder idea in their head.
	- Lines 355-361: Where does a classifier suddenly come from? How does it look like, how was it trained, and on what data?
	- Line 428: I still dislike "within the data's convex hull" since it is not the data $X$ but rather the embedding $A$. The reader might think about $X$ when reading "data".
	- Figure 6: Labels for the $x$ and $y$-axes are missing.
	- Figure 8: How was $A$ projected to two dimensions for the sake of this visualization (point cloud)? How were the points colored? What does the vertical band on the right-hand side with added transparency mean?
	- Equation (7): If $A=ZD$ as in line 955, then $D$ is a $k \times d$ matrix. There is no $n$, but $n$ is used in Equation (7).
	- Line 1091: Why $m$ and not $n'$?

**Questions For Authors:**

1. Why is archetypal analysis not a baseline?
2. Figures 2,3,4: What was $k$ in those experiments?
3. For the stability in Equation (2), the dictionary $D$ is assumed to reside on the sphere. Where and how is that done? Is $D$ always forced to be on the sphere or just normalized for computing the stability? Please clarify.
4. Lines 1109-1114 mention that $D$ is constrained to be on the sphere. This is not for A-SAE but rather an argument for standard SAEs, right?
5. Figure 3: Do all 5 points of A-SAE overlap? What does the dashed line indicate? Were all dictionaries normalized to lie on the sphere for the sake of computing the stability?
6. Line 302: You state that ImageNet has 1.2M points and make a similar comment in lines 1078-1082. Did you use any special variant of $K$-Means? How did you initialize $K$-Means?
7. Line 269: I suggest to remind the reader where $Z$ comes from for your A-SAE.
	- **Assuming that $Z$ is like in archetypal analysis**, i.e., $Z$ is not only non-negative but also row-stochastic (each row sums to one). This is correctly stated in lines 244-245: "representing each data point as a convex combination of Archetypes".
		- Lines 81-82: Archetypal analysis constrains the archetypes to lie **on the boundary of the convex hull** of data (exception: only one archetype)! See Proposition 1 in Cutler & Breiman (1994). Stating that the atoms are **within the convex hull** is thus wrong.
		- Lines 239-245: Please rephrase this. This geometric flexibility will reduce the reconstruction error of archetypal analysis drastically. However, if I perturb my data or retrain with random initialization, my latent space can be sufficiently different such that the location and the meaning of archetypes change as well.
		- Line 253: Again the issue with lying within the convex hull.
		- Line 268: The statement "each archetype originates from the data" is misleading. An archetype resides on the boundary of the convex hull of the data and an archetype does not have to be a data point, it is rather a convex combination of data.
	- **Assuming that $Z$ is as in Equation (1) (line 201), i.e., $Z \geq 0$**, then it should be mentioned clearly because only a few sentences before you state how archetypal analysis is doing it and everyone familiar with archetypal analysis reading this paper will be confused.
	- Can you clarify already now?
8. Line 291: "Models were trained on the ImageNet dataset" means that the five evaluated models were trained on ImageNet, but where were the SAE and NMF variants trained? Same dataset? Please clarify.
9. Line 302: "The data matrix $A$ was element-wise standardized." What does that mean?
10. Tables 1,2,3: Are the results from a single run or were they averaged (how, over how much repetitions, why no measure of spread like std or stderr)?
11. Table 1: Why is A-SAE missing and what was the parameter for RA-SAE? How was it chosen?
12. Figure 10: I wonder what the green line "Convex_hull" refers to. It could be the vertices of the convex hull of $A$ which yields $conv(A)=conv(C)$, thus the error would be as good as for $A$ and thus the best line. However, knowing about the complexity of computing convex hulls, I do not believe that the authors computed a convex hull on 1M data points in a space with more than a handful (25) dimensions.

**Relation To Broader Scientific Literature:**

- SAEs are extended by ideas from archetypal analysis.

**Theoretical Claims:**

- Proposition F.1: Those statements are rather obvious and known.

---

> ### Author Rebuttal · Authors · 2025-04-01
>
> We thank the reviewer for the detailed review—we sincerely appreciate the rigorous and constructive evaluation! The suggested related work and the thoughtful questions are very helpful. While not all references pertain directly to SAEs, the community we attempt to reach, the body of work highlighted on archetypal analysis is highly relevant. We have taken this review seriously and respond to specific comments below.
>
> ---
>
> > **On Related Work**
>
> Thank you for the extensive list of relevant references! These papers provide valuable context and inspiration on Archetypal analysis and we promise to add a detailed discussion of these papers in our related work in the final version of the paper. However, we do want to emphasize that **the goal of our paper is not to advance the field of archetypal analysis per se, but to bridge it with the neural network interpretability community by defining SAEs inspired by it.** To this end, we note none of the suggested papers address neural network explainability directly (and most do not even mention it). For the interpretability community, these geometric priors have not been integrated with modern large-scale SAE pipelines, and our contribution lies in highlighting such ideas for large models’ interpretability.
>
> Overall, while we stand on the strong theoretical shoulders of the Archetypal analysis community, we note our work serves a different audience: researchers in interpretability and applied representation learning who benefit from the stability, structure, and plausibility introduced by archetypal anchoring, even if they do not come from the convex analysis or matrix factorization literature.
>
> ---
>
> > **Clarifying Our Position Relative to Archetypal Analysis**
>
> Building on our response above, we would also like to clarify the precise relationship between our proposed Archetypal-SAE pipeline and Archetypal analysis more broadly. Our approach, Archetypal SAE, constrains the **dictionary** $D$ to be in the **convex hull of data**, i.e., $D = WA$ with $W \in \Omega_{k,n}$ (row-stochastic). However, the sparse code **Z is not constrained to be row-stochastic**, and hence we do not implement archetypal analysis in its classical manner; rather, we merely apply its geometric anchoring idea to the SAE decoder. Nevertheless, we call our proposed pipeline “Archetypal SAE” as an ode to the concept of Archetypal analysis, from which we derived the idea!
>
>
> ---
>
> > **On Convex Hull Approximation and Anchor Subset (C)**
>
> Please note we do not claim novelty in using K-Means for calculating a reduced subset of anchors. Our proposed Archetypal SAEs are in fact designed in a modular way, and we can expect better subset selection methods (e.g., your suggested ones) will only improve the results further! To this end, we also note here that we experimented with several viable methods when designing Archetypal-SAEs (e.g., Isolation Forest, One-Class SVM). In practice, we found K-Means to offer the best trade-off between **scalability, reconstruction, and stability**. Nevertheless, we believe your suggested methods will be exciting to experiment with in future!
>
>
> ---
>
> > **On the Stability Evaluation (Claim 1)**
>
> We emphasize we **do evaluate repeated SAE training runs** and report stability in **Figure 3** using the cosine alignment metric of Equation (2). Each point in our results is the mean over **four independently trained models with different seeds**, and the Hungarian matching is applied post hoc to align dictionary atoms. We will highlight this better in the final version of the paper!
>
> ---
>
> > **Clarifications and Revisions**
>
> We will revise the paper to address the points you've raised:
>
> - Fourth claim: Our dataset will be openly released upon acceptance.
> - Clarify that `data’ refers to **model activations**, not input images.
> - Make recommended notation changes (e.g., D vs D′, A vs A′), capitalization, etc.
> - Improve figure captions as recommended (e.g., stability computation in Fig. 3).
> - Clarify classifier usage in a plausibility benchmark (we use the backbone's own classification head).
> - Address missing definitions (e.g., Lambda, Z, etc.).
> - Explain dataset standardization: activation are taken after layer norm (without the affine part), thus each activation \(mean(a) = 0, std(a) = 1 \).
>
> - **Training dataset for SAEs:** We note all SAEs (and NMF variants) were trained on **ImageNet-1K activations**, using ∼250M tokens depending on architecture.
>
> - **Initialization of K-Means:** We used K-Means++ with mini-batch training for scalability.
>
> ---
> ---
>
> **Summary.** Thank you again for the time and care you put into your review. We believe that your feedback substantially improved the clarity, rigor, and positioning of our work. We will incorporate your suggestions into the final version, including expanded citations and clearer technical exposition.

---

> > ### Comment · Reviewer_hoxe · 2025-04-07
> >
> > Thank you very much for your clarifications.
> >
> > ---
> >
> > > **On Related Work**
> >
> > > However, we do want to emphasize that the goal of our paper is not to advance the field of archetypal analysis per se, but to bridge it with the neural network interpretability community by defining SAEs inspired by it.
> >
> > I agree. Nevertheless, it felt that the discussion around AA fell short, especially in the related work section (and in Appendix C). I appreciate that the authors are committed to improve upon that aspect.
> >
> > ---
> >
> > > **Clarifying Our Position Relative to Archetypal Analysis**
> >
> > Yes, seeing a comparison with vanilla AA would have been interesting nevertheless. However, this is not critical.
> >
> > Please clearly state the difference to vanilla AA once.
> >
> > ---
> >
> > > **On Convex Hull Approximation and Anchor Subset (C)**
> >
> > Using $k$-means to form $C$ will definitely shrink the volume, i.e., $\text{vol}(\text{conv}(C)) \leq \text{vol}(\text{conv}(X))$. I still believe that your relaxed version (not novel!) is just needed because of this. There are better (and faster) ways to construct $C$ which maintains the volume or at least shrinks it not as much as $k$-means. It would be great if you could test this at some time (not needed within this rebuttal discussion!).
> >
> > ---
> >
> > > **On the Stability Evaluation (Claim 1)**
> >
> > It seems that I have missed that. I update the corresponding part of the review.
> >
> > Did you consider showing horizontal and vertical lines (plotted with transparency) showing, e.g., a standard deviation to visualize the spread? Just a thought..
> >
> > ---
> >
> > > **Clarifications and Revisions**
> >
> > Thank you!
> >
> > ---
> >
> > If you still have time to reply, do you mind answering my questions 2, 3, 4, 10, 11, 12?
> >
> > ---
> >
> > I appreciate that the authors are committed to improve their paper. I reconsidered my evaluation, updated parts of my review (reflecting the current status, not the promised version of the paper), and raised my score.

---

> > > ### Author Response · Authors · 2025-04-09
> > >
> > > Thank you for the thoughtful follow-up and for raising your score -- **we truly appreciate your detailed and constructive engagement**.
> > >
> > > We agree that the related work discussion on archetypal analysis deserves more depth, and we’re committed to expanding it meaningfully in the final version. Your remarks on convex hull volume and anchor subset selection are well taken; while we focused on K-Means for scalability, we fully agree that stronger subset selection could reduce the need for relaxation, and we’re excited to explore those directions further.
> > >
> > > On the rebuttal: we initialy had a longer response covering all your (great) questions, but unfortunately had to trim it significantly to comply with the character limit.  We prioritized addressing a core subset -- in depth -- while committing to integrate the rest into the final version of the paper.
> > >
> > > To quickly follow up on your numbered questions:
> > >
> > > - (2) We used `k = 5d`, where `d` is the feature dimension of the backbone (e.g., 768 for ViT-B) as indicated in the Setup section.
> > > - (3) For stability evaluation, each row of `D` are l2-normalized post hoc to allow a fair comparison. This normalization is only used at evaluation time and applies to all methods equally.
> > > - (4) Correct.
> > > - (10) Each activation vector in the matrix `A` is taken after the model’s LayerNorm, meaning activations typically have zero mean and unit variance. However, some backbones include a LayerNorm with an affine transformation. To remain faithful to the model’s internal representations, we preserved the affine component when present. We’ve added a note in the setup section to clarify this behavior (see also: https://pytorch.org/docs/stable/generated/torch.nn.LayerNorm.html).
> > > - (11) All results are averages over 4 independent runs with different random seeds.
> > > - (12) A-SAE corresponds to $\delta=0$ and appears as a special case in the RA-SAE ablation (Fig. 4). It underperforms in reconstruction compared to relaxed variants, so we focused on RA-SAE. We chose $\delta = 0.01$ as a default via ablation.
> > >
> > >
> > > Thanks again for your rigorous and generous review. Your input substantially improved the work.

---

### Official Review · Reviewer_27og · 2025-03-12

**Overall Recommendation:** 4

**Summary:**

The paper proposes an extension of vanilla SAE approaches to archetypal SAE, a type of geometric anchoring that improves various shortcomings, stability and plausibility, of vanilla SAEs. Further, the authors introduce two new benchmarks for plausibility and identifiability. The paper thoroughly evaluates the contributions across a variety of SOTA feature extractors and demonstrates its effectiveness.

**Claims And Evidence:**

- The authors claim the current SAE approaches are instable and do a good job of showcasing this with experiments

**Essential References Not Discussed:**

None

**Experimental Designs Or Analyses:**

- The introduced relaxation parameter is intuitive and well ablated
- In the setup section, the authors claim all models were trained on IN1K. I guess this is referring to their own tuning of the models, but it’s written a bit confusing. Please make it more clear this you are finetuning with IN1K. At the moment, it sounds like all models were trained on IN1K initially.
- The quantitative evaluations are extensive and highlight the proposed method well
- The paper starts with a qualitative teaser. However, the main evaluation section completely lacks a qualitative evaluation. It would be great to add that.
- The proposed benchmarks are an interesting addition to the previous analysis

**Methods And Evaluation Criteria:**

- The paper in general and method description specifically is well structured and easy to follow
- The formulation of the stability measure makes a lot of sense
- The aspect of scalability is important and the proposed relaxation tackles this issue well

**Other Comments Or Suggestions:**

Since my knowledge is this space is limited, I will take into account the judgement from the other reviewers to make a final decision

**Other Strengths And Weaknesses:**

Strengths:
- The paper  is well written and easy to follow
- The stability measure makes sense and is well formulared
- Its Great that the aspect of scalability is investigates for the method and the proposed relaxation achieves this property
- The benchmark contributions are relevant to the Community

Weaknesses:
- The paper lacks qualitative evaluations in the Main text
- The writing of what models were trained on what datasets is confusing
- Only contrastive and vision language models were evaluated. Would be Great to also see masked image modeling approaches

**Questions For Authors:**

- The Experiments section only evaluates Contrastive, Vision-Language and supervised approaches. I think what would it be interesting to know is how good MIM-based approaches are like Masked Autoencoders [1] or iBOT [2]? I think they are compatible with timm
- Please clear up the confusion about model training in the setup section
- Would you be able to squeeze in qualitative results into the main evaluation section?

[1] He, Kaiming, et al. "Masked autoencoders are scalable vision learners." Proceedings of the IEEE/CVF conference on computer vision and pattern recognition. 2022.
[2] Zhou, Jinghao, et al. "Image BERT Pre-training with Online Tokenizer." International Conference on Learning Representations.

**Relation To Broader Scientific Literature:**

The authors do a good job to put into perspective the previous approaches incl. SAEs and how they tackle these issues

**Theoretical Claims:**

- The claims about limitations of standard SAEs in L270ff are backed up with a reference

---

> ### Author Rebuttal · Authors · 2025-04-01
>
> Thank you for your thoughtful and constructive review! We appreciate your recognition of the paper’s strengths, including the methodological clarity, the novel evaluation metrics, and the thorough empirical analysis. Below, we address specific comments.
>
> ---
>
> > **Clarification of Dataset Usage (Setup Section)**
>
> Good point! Based on our current phrasing of the setup, a reader may mistakenly conclude that the *vision backbones* were trained / fine-tuned on IN1K. This is not the case—the backbones (e.g., DINOv2, SigLip) were in fact pretrained models sourced from the 'timm' library. We specifically used IN1K to subsequently train the SAEs analyzed in this paper, computing activations from the backbone models, regardless of their pretraining method, for this purpose. We will ensure this is made clear in the final manuscript—thank you!
>
> ---
>
> > **Inclusion of Qualitative Results in the Main Evaluation**
>
> We strongly agree that our qualitative results, as teased in Figure 1, would be useful to include in the main paper! Unfortunately, space constraints forced us to defer our in-depth qualitative analysis to the appendix. In brief, we note that in **Figure 7** we showed "exotic concepts" discovered by RA-SAE on DINOv2, such as a 'barber' concept that uniquely activates for barbers (not clients), fine-grained concepts based on petal edges, and a shadow-based feature suggestive of 3D reasoning. Similarly, in **Figure 8** we highlighted RA-SAE’s ability to disambiguate a single ImageNet class (e.g., rabbit) into spatially localized subcomponents and fine-grained animal facial features, while in **Figure 10** we reveal emergent *concept clusters*, such as spatial-relational concepts (“underneath”) and fine-grained animal facial features.
>
> In response to your comment, we promise to use the extra page provided for final manuscripts to pull a subset of these results back to the main paper.
>
> ---
>
> > **On backbone pretraining approaches, e.g., Masked Image Modeling (MIM)**
>
> We emphasize that we already analyze DINOv2 models, which are pretrained using iBOT, an MIM-based objective. That is, while we did not include a standalone MAE model, our experiments do include an MIM-trained model. We will clarify this point in the revised text by providing further details of the pipeline used for training backbone models analyzed in our work, hence also making explicit that we already analyze a backbone trained via MIM pretraining.
>
> ---
> ---
>
> **Summary:** Thank you again for the constructive feedback! We hope our responses help address your raised questions, and, in line with your suggestion, we promise to pull back qualitative results from the appendix to further improve the main paper. Given these changes, we would be grateful if you continue to champion our paper's acceptance!

---

> > ### Comment · Reviewer_27og · 2025-04-08
> >
> > Thank you to the authors for considering my concerns. Overall, I believe the paper is in good shape! I will raise my score

---

> > > ### Author Response · Authors · 2025-04-09
> > >
> > > Thank you for your thoughtful feedback and for revisiting your score.
> > >
> > > We're glad the clarifications and additional qualitative results addressed your concerns, and we truly appreciate your recognition of the paper’s contributions. Thanks again for your support in championing this work.

---

### Decision · Program_Chairs · 2025-05-01

**Decision:**

Accept (poster)

**Comment:**

**Strengths:**

(1) This paper tackles the instability problem of sparse autoencoders and proposes a principled solution via archetypal constraints.

(2) Extensive experiments demonstrate improved stability, structure, and plausibility of the learned concepts across multiple vision backbones (e.g., DINOv2, ConvNeXt, SigLIP), which are supported by well motivated benchmarks for plausibility and identifiability.

**Weaknesses:**

(1) The paper does not evaluate on LLMs or vision-language models in the spirit of interpretability (beyond SigLIP), which is limiting the generality of the proposed method.

(2) The proposed stability metric is innovative, but its connection to interpretability is not convincingly established.

**Discussion:**

There is general agreement among reviewers that the paper addresses an important and under-explored challenge in interpretable representation learning -- stability of concept extraction via sparse autoencoders. While reviewers 27og and hoxe upgraded their evaluations after rebuttal, reviewer 1Jmo remained skeptical about propose evaluation metrics, and reviewer DiCd held reservations about the paper’s applicability to more complex, contemporary architectures (e.g., LLMs, VLMs). That said, all reviewers acknowledged the paper’s strengths, and most concerns were well-addressed in the rebuttal.

**Recommendation: Accept**

Despite some open questions about evaluation framing and scope of generalization, the paper offers a clear and well validated contribution with promising implications for concept-based interpretability. Its improvements in stability, coupled with strong empirical results, make it a valuable addition to the ML community.